# Bayesian Tensor Decomposition with Diffusion Model Prior

**Zerui Tao** [1]   **Qibin Zhao** [1]

## Abstract

Low-rank tensor decomposition (TD) is usually effective on clean, fully observed data, but it often degrades under severe missingness or noise. Low-rankness is itself a useful but limited *structural* prior, and additional handcrafted priors (*e.g.*, sparsity or smoothness) still fall short of capturing the rich statistics of real-world data. To compensate for this weak inductive bias under heavy corruption, one would like to inject a *learned*, data-driven prior; however, the state-of-the-art diffusion models are not readily compatible with current TD and tractable posterior inference. To address these challenges, we introduce DiffBCP, a *hybrid-prior* Bayesian CP decomposition framework that couples a cumulative shrinkage process prior over the CP factors for automatic rank selection with an off-the-shelf pre-trained diffusion model as an implicit data prior on the reconstructed tensor. To make posterior inference tractable despite the coupling among the likelihood, low-rank constraint, and diffusion prior, we develop a split Gibbs sampler: CP factors admit conjugate updates, while the diffusion block is sampled via low-rank-guided denoising. A noise-adaptive coupling schedule further reduces sensitivity to hand-tuned annealing. Experiments on image inpainting and denoising, including high-resolution out-of-distribution images, show consistent gains over Bayesian, nonlinear, and plug-and-play TD baselines.

## 1. Introduction

Tensor decomposition (TD) is a powerful tool for multi-dimensional data analysis, with applications in machine learning, signal processing, and scientific computing (Kolda & Bader, 2009; Cichocki et al., 2016). By factorizing high-order tensors into contractions of much smaller factors, TD enables efficient representation, compression, and interpretation of complex data structures. One important application is signal recovery from incomplete and noisy observations, where TD can exploit the low-rank structure of the underlying data to reconstruct missing or corrupted entries (Sidiropoulos et al., 2017).

In the research field of TD, one scheme is to study effective decomposition structures to incorporate prior beliefs about the data latent representations, such as CP (Hitchcock, 1927), Tucker (Tucker, 1966), tensor networks (Oseledets, 2011; Zhao et al., 2016), and many other decompositions (Kolda & Bader, 2009; Kilmer & Martin, 2011; Cichocki et al., 2016). The motivation behind is that when the model structure matches the data hidden pattern better, the TD method can achieve better representation and recovery performances. Motivated by this, efforts have been made to design more flexible TD models that can learn richer structures from the data, from traditional multi-linear models to non-linear models (Zhe et al., 2016; Liu et al., 2019).

Ironically, in real applications such as image processing, we usually observe that, when the data are complete and clean, even the simplest TD structure like CP decomposition achieves strong performances; on the contrary, when the data are partially observed or noisy, more sophisticated TD models may not lead to satisfactory results. The main challenge is that the low-rank assumption, while itself an effective *structural* prior, becomes insufficient on its own once the observations are noisy or incomplete. To reduce this gap, previous work augments TD with additional handcrafted priors such as sparsity or smoothness (Sun et al., 2017; Wang et al., 2017), but these handcrafted priors still may not fully exploit the rich information present in real-world data. This raises a natural question: *Can we complement the structural low-rank prior with a more powerful, learned data prior for noisy and incomplete data?*

To address this issue, we adopt a Bayesian framework that can naturally couple a handcrafted structural prior with a learned, data-driven prior in a single probabilistic model. In particular, we propose DiffBCP, a hybrid-prior probabilistic CP decomposition model that leverages the recent advances in diffusion models (Karras et al., 2022). Con-

[1]RIKEN Center for Advanced Intelligence Project (AIP), Tokyo, Japan. Correspondence to: Qibin Zhao <qibin.zhao@riken.jp>.

*Proceedings of the 43rd International Conference on Machine Learning*, Seoul, South Korea. PMLR 306, 2026. Copyright 2026 by the author(s).

cretely, a powerful pre-trained diffusion model is placed in the Bayesian CP model as an implicit prior on the reconstructed tensor, enabling the model to capture complex data distributions effectively. To ensure low-rank structure and automatic rank determination, we adopt the cumulative shrinkage process (CUSP) prior (Legramanti et al., 2020) on the CP component weights. The resulting model combines the strengths of Bayesian tensor decomposition and diffusion models, allowing for robust learning from noisy and partially observed tensors. To sample from the posterior distribution, we adopt the split Gibbs sampler (SGS, Vono et al., 2019) with an auxiliary variable, which decouples the likelihood, the implicit diffusion prior, and the low-rank constraint. This leads to closed-form updates for the latent factors and conjugate sampling. The diffusion prior block is sampled via low-rank–guided denoising, turning plug-and-play guidance into a structured Bayesian inference procedure. While the noise level is an important factor in the SGS, our fully Bayesian framework allows us to infer it automatically from the data, eliminating the need for manual tuning. We conduct experiments on image inpainting and denoising tasks. Results show that our method performs well on various types of images, demonstrating the effectiveness of combining diffusion model priors with Bayesian tensor decomposition.

## 1.1. Contributions

Our main contributions are summarized as follows:

- We propose a *hybrid-prior* Bayesian CP model with the sparse CUSP prior on the latent factors and powerful pre-trained diffusion model prior on the reconstructed tensor to capture complex data distributions.
- The inference is fully probabilistic, giving better noise adaptation and annealing ability.
- The proposed method achieves superior performance on image datasets, even outperforms non-linear TDs.

## 1.2. Related Work

**Bayesian tensor decomposition** Bayesian TD has been studied for various forms of decompositions, including CP (Zhao et al., 2015; Rai et al., 2014; Cheng et al., 2022), Tucker (Schein et al., 2016; Fang et al., 2021; Stolf & Canale, 2025), tensor train/ring (Long et al., 2021; Tao et al., 2023a; Xu et al., 2023), and non-parametric TD (Zhe et al., 2016; Tao et al., 2024). Previous work mainly focuses on developing better priors on the latent factors, for example sparsity-inducing priors (Zhou et al., 2015; Tillinghast et al., 2022) or smoothness priors (Chen & Sun, 2022). In particular, Stolf & Canale (2025) develop a Bayesian Tucker using a similar CUSP prior on Tucker factor matrices. Other methods learn more flexible likelihood models, such as mixture-of-Gaussians (Chen et al., 2016) and deep

generative models (Tao et al., 2023b; Chen et al., 2025). However, these works do not consider powerful data-driven priors on the reconstructed tensor itself.

**Generative priors for inverse problems** Generative priors for general inverse problems have a long history, ranging from plug-and-play denoisers (Venkatakrishnan et al., 2013) and regularization-by-denoising (Romano et al., 2017) to generator-based priors such as CSGM (Bora et al., 2017) and Deep Image Prior (Ulyanov et al., 2018). The rise of diffusion models has produced much stronger natural-image priors, leading to a family of diffusion posterior samplers including DPS (Chung et al., 2023), SNIPS/DDRM (Kawar et al., 2021; 2022), and split-Gibbs-style samplers (Wu et al., 2024; Xu & Chi, 2024; Heurtel-Depeiges et al., 2024), with the recent benchmark of Zheng et al. (2025). In the TD literature, Zhao et al. (2020; 2022) pair CNN denoiser priors with ADMM, but the denoiser is much less expressive than a diffusion model and the framework is non-probabilistic.

DiffBCP sits at the intersection of these two lines: built on the split Gibbs sampler (Vono et al., 2019; Wu et al., 2024), it additionally imposes a Bayesian low-rank CP structure with CUSP shrinkage on the factor weights, yielding a *hybrid-prior* formulation within a fully probabilistic framework.

## 2. Preliminaries

**Notations** We use bold lowercase letter, capital letters, and caligraphical capital letters to denote vectors, matrices, and tensors respectively, *e.g.*, $\mathbf{x} \in \mathbb{R}^I, \mathbf{X} \in \mathbb{R}^{I_1 \times I_2}$, and $\mathcal{X} \in \mathbb{R}^{I_1 \times I_2 \cdots \times I_N}$. Subscripts represent slices of sub-arrays, *e.g.*, $\mathbf{X}_{i::}$ is the $i$-th slice of $\mathcal{X}$ along the first dimension. The mode-$n$ matricization is denoted as $\mathbf{X}_{[n]}$, where $\mathbf{x}_{[n],i_n:} = \text{vec}\left(\mathcal{X}_{\cdots i_n \cdots}\right), \forall i_n = 1, \ldots, I_n$. We use underlined vectors to represent the set of indices, *e.g.*, $\underline{\mathbf{i}} = (i_1, i_2, \ldots, i_N)$. We use $\circ$ to denote outer product, $\odot$ to denote Khatri-Rao product, and $*$ to denote Hadamard product. $D_{\text{TV}}$ and $D_{\text{KL}}$ denote the total variation and Kullback-Leibler divergence.

### 2.1. Tensor Decomposition

The goal of tensor decomposition is to learn parsimonious representations of multi-dimensional arrays, *e.g.*, $\mathcal{X} \in \mathbb{R}^{I_1 \times I_2 \times \cdots \times I_N}$, where $N$ is the order of the tensor and $I_n$ is the dimension along the $n$-th mode. In this work, we focus on the CP decomposition, denoted as,

$$\mathcal{X} = \text{CP}\left(\boldsymbol{\lambda}, \mathbf{A}^{(1)}, \mathbf{A}^{(2)}, \ldots, \mathbf{A}^{(N)}\right) \quad (1)$$
$$= \sum_{r=1}^{R} \lambda_r \mathbf{a}_{:r}^{(1)} \circ \mathbf{a}_{:r}^{(2)} \cdots \circ \mathbf{a}_{:r}^{(N)},$$

where the number $R$ is the CP rank. The CP form factorizes a tensor into a sum of rank-one tensors. $\mathbf{A}^{(n)} \in$

$\mathbb{R}^{I_n \times R}, \forall n = 1, \ldots, N$ are called factor matrices, and $\boldsymbol{\lambda} \in \mathbb{R}^R$ is the weight vector. Usually, TD problems are solved by minimizing the loss functions such as,

$$\min_{\boldsymbol{\lambda}, \mathbf{A}^{(1)}, \ldots, \mathbf{A}^{(N)}} \left\| \mathcal{O} * \left( \mathcal{Y} - \mathsf{CP}\left( \boldsymbol{\lambda}, \mathbf{A}^{(1)}, \ldots, \mathbf{A}^{(N)} \right) \right) \right\|_F^2,$$

where $\mathcal{Y}$ is the observation and $\mathcal{O}$ is the binary mask. Usually, adding prior information or regularization on the latent factors $\mathbf{A}^{(1:N)}$ or the estimate tensor $\mathcal{X}$ can improve the performance. However, these priors are usually heuristic and handcrafted such as smoothness (Chen & Sun, 2022) or sparsity (Zhou et al., 2015; Tillinghast et al., 2022).

## 2.2. Diffusion Model

Diffusion models have become the most powerful class of generative models, especially for images. Diffusion models have been proposed from different disciplines (Sohl-Dickstein et al., 2015; Ho et al., 2020; Song et al., 2021). The work of EDM (Karras et al., 2022) provides a unified framework to understand different diffusion models. Suppose the data is diffused by $\mathbf{x}_t = s(t)\mathbf{x} + \sigma(t)\boldsymbol{\epsilon}$, where $s(t)$ and $\sigma(t)$ are two functions of time $t$, and $\boldsymbol{\epsilon} \sim \mathcal{N}(\mathbf{0}, \mathbf{I})$. Denote $p(\mathbf{x}_t; \sigma(t))$ as the diffused distribution. The reversed denoising process follows the SDE,

$$\mathrm{d}\mathbf{x}_t = \left[ u(t)\mathbf{x}_t - v(t)\nabla_{\mathbf{x}_t} \log p\left( \frac{\mathbf{x}_t}{s(t)}; \sigma(t) \right) \right] \mathrm{d}t + \sqrt{v(t)} \, \mathrm{d}\bar{\mathbf{w}}_t, \quad (2)$$

where $u(t) = \dot{s}(t)/s(t)$, $v(t) = 2s(t)^2 \dot{\sigma}(t)\sigma(t)$, and $\bar{\mathbf{w}}$ is the standard Wiener process. The only unknown term is the score function $\nabla_{\mathbf{x}} \log p(\mathbf{x}; \sigma(t))$, which is approximated by a neural network $s_\psi(\mathbf{x}_t, t)$ and trained by score matching techniques. By solving the SDE from a large $t$ to 0, we can sample from the data distribution or denoise noisy data. Nowadays, diffusion models have become the state-of-the-art generative models, and there are many off-the-shelf pre-trained diffusion models for different types of data.

# 3. Proposed Model and Algorithm

## 3.1. Probabilistic CP Decomposition

To incorporate powerful data priors into the CP decomposition, we develop the probabilistic CP decomposition model. The joint probability of the proposed decomposition is,

$$p(\mathcal{Y}, \mathcal{X}, \boldsymbol{\lambda}, \mathbf{A}^{(1:N)}, \tau) \propto$$
$$p(\mathcal{Y} \mid \mathcal{X}, \tau)\, p(\mathcal{X} \mid \mathbf{A}^{(1:N)}, \boldsymbol{\lambda})\, p(\mathbf{A}^{(1:N)})\, p(\boldsymbol{\lambda})\, p(\mathcal{X})\, p(\tau),$$
$$(3)$$

where $\boldsymbol{\lambda}, \mathbf{A}^{(1:N)}$ are the CP factors in Eq. (1) and $\mathcal{X}$ is the reconstructed tensor. The structural factor $p(\mathcal{X} \mid \mathbf{A}^{(1:N)}, \boldsymbol{\lambda}) = \delta(\mathcal{X} - \mathsf{CP}\left( \boldsymbol{\lambda}, \mathbf{A}^{(1)}, \mathbf{A}^{(2)}, \ldots, \mathbf{A}^{(N)} \right))$ encodes the low-rank CP constraint; $p(\mathbf{A}^{(1:N)})$, $p(\boldsymbol{\lambda})$, and

$p(\tau)$ are the conjugate factor priors specified below; and $p(\mathcal{X})$ is a powerful pre-trained diffusion prior placed directly on the reconstructed tensor. Although $\mathcal{X}$ enters the joint through two factors, they play complementary roles: $p(\mathcal{X} \mid \mathbf{A}^{(1:N)}, \boldsymbol{\lambda})$ is a hard structural constraint that ties $\mathcal{X}$ to the CP factors, while $p(\mathcal{X})$ is a soft data-driven regularizer on the reconstructed tensor. The two priors $p(\mathcal{X})$ and $p(\mathbf{A}^{(1:N)})p(\boldsymbol{\lambda})$ jointly form a *hybrid prior* that fuses a structural low-rank component with a learned data-driven component in a single posterior. To our knowledge, this is the first Bayesian tensor decomposition that admits an off-the-shelf diffusion model as the data prior, extending the plug-and-play paradigm (Venkatakrishnan et al., 2013; Romano et al., 2017; Wu et al., 2024) from generic inverse problems to a fully probabilistic CP model with automatic rank and noise inference. For the likelihood, we consider each element is independently observed from the Normal distribution,

$$y_{\underline{\mathbf{i}}} \mid x_{\underline{\mathbf{i}}}, \tau \sim \mathrm{Normal}(y_{\underline{\mathbf{i}}} \mid x_{\underline{\mathbf{i}}}, \tau^{-1}), \quad \forall \underline{\mathbf{i}} \in \Omega,$$

where $\Omega$ denotes the set of observed entries in $\mathcal{O}$.

**Latent variable priors** Let $\{\tau, \boldsymbol{\lambda}, \mathbf{A}^{(1:N)}\}$ be latent variables. The precision follows a conjugate Gamma prior,

$$\tau \sim \mathrm{Gamma}(\tau \mid \alpha_0, \kappa_0),$$

where $\alpha_0$, $\kappa_0$ are the shape and rate parameters respectively. For the weight, we adopt the CUmulative Shrinkage Process (CUSP) prior (Legramanti et al., 2020), which provides column-wise shrinkage that, combined with the rank-adaptation procedure below, makes DiffBCP adaptive to unknown CP rank,

$$\lambda_r \mid \theta_r \sim \mathcal{N}(0, \theta_r), \ \theta_r \mid \pi_r \sim (1 - \pi_r)P_0 + \pi_r \delta_{\theta_\infty},$$
$$\pi_r = \sum_{l=1}^{r} \omega_l, \quad \omega_l = \nu_l \prod_{m=1}^{l-1} (1 - \nu_m),$$

where $\nu_l \overset{\text{iid}}{\sim} \mathrm{Beta}(1, \beta)$, $\theta_\infty$ is a small positive constant, $\delta_{\theta_\infty}$ is the Dirac delta function, and $P_0$ is a starting continuous distribution. As the rank $r$ increases, the distribution of $\lambda_r$ gradually diffuses from $P_0$ to $\delta_{\theta_\infty}$ almost surely. This is similar to a spike-and-slab prior, but with hierachical mixture proability that can increasingly shrink factors. In practice, we set $P_0 = \mathrm{InvGamma}(\alpha_\theta, \beta_\theta)$ with $\alpha_\theta > 1$ and $\theta_\infty = 1\mathrm{e}{-}3$. For the factor matrices, we assume each element follows a Gaussian prior,

$$a_{ir}^{(n)} \sim \mathrm{Normal}(0, 1), \quad \forall n, i, r.$$

This prior regularizes the norm of factor matrices, so that we can truncate the rank based on the value of $\boldsymbol{\lambda}$.

**Diffusion model prior** In the whole probabilistic model in Eq. (3), we still need to specify the data prior $p(\mathcal{X})$. Most existing probabilistic TD methods leave $p(\mathcal{X})$ implicit or treat it as a non-informative constant, in which case Eq. (3) reduces to a classical Bayesian CP (Rai et al., 2014; Zhao et al., 2015) (up to the CUSP shrinkage on the weights) and the model only exploits low-rank structure. We instead instantiate $p(\mathcal{X})$ as a pre-trained diffusion model, which contributes a learned natural-data expert on the reconstructed tensor and is the key ingredient that allows DiffBCP to recover rich textures and structures under heavy missingness and noise. We assume the reconstructed tensor $\mathcal{X}$ follows a diffusion model prior, that means, the diffused proabability follows,

$$\nabla_{\mathcal{X}_t} \log p(\mathcal{X}_t; \sigma(t)) = s_\psi(\mathcal{X}_t, t),$$

where $s_\psi$ is a pre-trained score network. Given this prior, we can sample $\mathcal{X}$ by solving the SDE in Eq. (2).

Considering the above probability formulation, we show that the CUSP prior can effectively shrink the residual of the CP decomposition.

**Theorem 3.1.** *Denote the $\underline{\mathbf{i}}$-th element in the $r$-th component as $x_{r,\underline{\mathbf{i}}} = \lambda_r \prod_{n=1}^N a_{i_n r}^{(n)}$. Suppose the factor is bounded by $\eta$ in second moment, i.e., $\mathbb{E}(a_{i_n r}^{(n)})^2 < \eta_n, \forall n, i_n, r$. Then $\forall \epsilon > 0$, we have,*

$$p((x_{r,\underline{\mathbf{i}}})^2 \geq \epsilon) \leq$$

$$\frac{1}{\epsilon} \left[ \left( \frac{\beta}{1+\beta} \right)^r \left( \frac{\beta_\theta}{\alpha_\theta - 1} - \theta_\infty \right) + \theta_\infty \right] \prod_{n=1}^N \eta_n.$$

*Remark* 3.2. The proof is deferred to appendix Section B.1. Since $\theta_\infty$ is usually set to a small value, the CUSP prior effectively shrinks the tail probability of each component as the rank $r$ increases. We can constrol the tail probability by adjusting the hyperparameter $\beta$. If we desire larger shrinkage, we may set a smaller $\beta$.

### 3.2. Split Gibbs Sampler

To get posterior samples, we need to sample from the joint distribution,

$$\exp(-\ell(\mathcal{Y}; \mathcal{X}, \tau) - g(\mathcal{X}) -$$
$$f_\lambda(\boldsymbol{\lambda} \mid \boldsymbol{\theta}) - f_{\boldsymbol{\theta}}(\boldsymbol{\theta}) - f_{\mathbf{A}}(\mathbf{A}) - f_\tau(\tau)), \quad (4)$$
$$s.t. \quad \mathcal{X} = \mathsf{CP}\left( \boldsymbol{\lambda}, \mathbf{A}^{(1)}, \mathbf{A}^{(2)}, \dots, \mathbf{A}^{(N)} \right).$$

Directly sampling from this distribution is difficult, because the likelihood, the prior on $\mathcal{X}$, and the prior of factors $\mathbf{A}$ are coupled together. Moreover, since the diffusion prior on $\mathcal{X}$ is implicit, even Langevin samplers for $\mathbf{A}$ are infeasible to implement.

To solve this problem, we adopt the Split Gibbs Sampler (SGS, Vono et al., 2019). Introducing an augmented variable $\mathcal{Z}$, the sampling problem becomes,

$$\exp(-\ell(\mathcal{Y}; \mathcal{X}, \tau) - g(\mathcal{Z}) - \phi(\mathcal{Z}, \mathcal{X}; \rho) -$$
$$f_\lambda(\boldsymbol{\lambda} \mid \boldsymbol{\theta}) - f_{\boldsymbol{\theta}}(\boldsymbol{\theta}) - f_{\mathbf{A}}(\mathbf{A}) - f_\tau(\tau)),$$
$$s.t. \quad \mathcal{X} = \mathsf{CP}\left( \boldsymbol{\lambda}, \mathbf{A}^{(1)}, \mathbf{A}^{(2)}, \dots, \mathbf{A}^{(N)} \right),$$

where $\phi$ is some distance measure and $\rho > 0$ is a hyperparameter controlling the coupling strength. If this distance measure converges to zero when $\mathcal{X}$ and $\mathcal{Z}$ are close, the marginal distribution converges to the original distribution (Vono et al., 2019). Then, we can derive the Gibbs sampler of each random variable. Following Vono et al. (2019); Wu et al. (2024), the distance is defined as,

$$\phi(\mathcal{Z}, \mathcal{X}; \rho) = \frac{1}{2\rho^2} \|\mathcal{Z} - \mathcal{X}\|_F^2. \quad (5)$$

#### 3.2.1. LATENT VARIABLE STEPS

First, we show how to sample all latent variables including the CP weight, factor matrices, the noise precision, and CUSP parameters. To sample from the CUSP prior, we introduce an augmentation variable $\zeta \in \{0, 1\}$.

**Sample $\boldsymbol{\lambda}$** For $r = 1, \dots, R$,

$$\lambda_r \mid \cdots \sim \mathcal{N}(\mu_r, \sigma_r), \quad (6)$$

where,

$$\sigma_r = \left( \theta_r + \tau \|\mathcal{O} * \mathcal{C}^r\|_F^2 + \frac{1}{\rho^2} \|\mathcal{C}^r\|_F^2 \right)^{-1}, \quad \mu_r =$$
$$\sigma_r \mathsf{Sum}\left( \tau \mathcal{O} * \mathcal{C}^r * (\mathcal{Y} - \mathcal{D}^r) + \frac{1}{\rho^2} \mathcal{C}^r * (\mathcal{Z} - \mathcal{D}^r) \right),$$

with $\mathcal{C}^r = \mathbf{a}_{:r}^{(1)} \circ \mathbf{a}_{:r}^{(2)} \circ \cdots \circ \mathbf{a}_{:r}^{(N)}$, $\mathcal{D}^r = \sum_{j \neq r} \lambda_j \mathcal{C}^j$, and $\mathsf{Sum}(\cdot)$ the sum of all elements.

**Sample $\mathbf{A}^{(n)}$** For $n = 1, \dots, N$ and $i = 1, \dots, I_n$, the full conditional distribution of $\mathbf{a}_{i:}^{(n)}$ is,

$$\mathbf{a}_i^{(n)} \mid \cdots \sim \mathcal{N}(\boldsymbol{\mu}_i^{(n)}, \boldsymbol{\Sigma}^{(n)}), \quad (7)$$

where,

$$\boldsymbol{\Sigma}^{(n)} = \left( \mathbf{I} + \mathbf{B}^{(n)} \mathrm{diag}(\tau \mathbf{o}_{[n],i:} + 1/\rho^2) \mathbf{B}^{(n)\mathsf{T}} \right)^{-1},$$
$$\boldsymbol{\mu}_i^{(n)} = \boldsymbol{\Sigma}^{(n)} \left( \tau \mathbf{y}_{[n],i:} \mathrm{diag}(\mathbf{o}_{[n],i:}) + \frac{1}{\rho^2} \mathbf{z}_{[n],i:} \right) \mathbf{B}^{(n)\mathsf{T}}.$$

with $\mathbf{B}^{(n)}$ defined by,

$$\mathbf{B}^{(n)} = \boldsymbol{\Lambda} \left( \mathbf{A}^{(N)} \odot \cdots \mathbf{A}^{(n+1)} \odot \mathbf{A}^{(n-1)} \odot \cdots \mathbf{A}^{(1)} \right)^{\mathsf{T}}.$$

*Remark* 3.3. We follow Rai et al. (2014) to adaptively adjust the number of components in $\boldsymbol{\lambda}$ (the CP rank). This procedure is denoted as adapt_factor($\boldsymbol{\lambda}, \mathbf{A}$) and details are shown in the appendix Section A. At sampling step $i$, we adapt the rank with probability $p(i) = \exp(\alpha_0 + \alpha_1 i)$, where $\alpha_0$ and $\alpha_1$ are hyperparameters. If there are inactive factors, we remove them. If all factors are active, we add a new factor (and associated parameters) from the priors. Together with the CUSP prior, this yields a *bidirectional* rank-adaptation scheme: CUSP shrinks redundant components toward $\delta_{\theta_\infty}$ which are then physically pruned, while a new component is sampled whenever the current rank turns out to be insufficient. Consequently, DiffBCP is robust to mis-specification of the initial rank in either direction.

**Sample $\tau$**   The noise precision can be sampled from a conjugate Gamma distribution,

$$\tau \mid \cdots \sim \text{Gamma}\left(\alpha_0 + \frac{|\Omega|}{2}, \kappa_0 + \frac{1}{2}\sum_{\underline{\mathbf{i}}\in\Omega}(y_{\underline{\mathbf{i}}} - x_{\underline{\mathbf{i}}})^2\right),$$
(8)

where $|\Omega|$ is the number of observed entries.

**Sample $\zeta$**   For $r = 1, \ldots, R$, the posterior $\zeta_r = h \mid \cdots \propto$

$$\omega_h \cdot \begin{cases} \text{Normal}(\lambda_r; 0, \theta_\infty), & h \leq r \\ \text{Student-t}_{2\alpha_\theta}(\lambda_r; 0, \beta_\theta/\alpha_\theta), & h > r. \end{cases}$$
(9)

**Sample $\nu$**   For $r = 1, \ldots, R-1$, the posterior $\nu_r \mid \cdots \sim$

$$\text{Beta}\left(1 + \sum_{h=1}^{R}\mathbb{1}(\zeta_h = r), \beta + \sum_{h=1}^{R}\mathbb{1}(\zeta_h > r)\right), \quad (10)$$

and set $\nu_R = 1$.

**Sample $\theta$**   For $r = 1, \ldots, R$, the posterior $\theta_r \mid \cdots \sim$

$$\text{InvGamma}\left(\alpha_\theta + \frac{1}{2}, \beta_\theta + \frac{\lambda_r^2}{2}\right)^{\mathbb{1}(\zeta_r > r)} \cdot \delta_{\theta_\infty}^{\mathbb{1}(\zeta_r \leq r)}. \quad (11)$$

### 3.2.2. AUGMENTED VARIABLE STEP

Second, we show how to sample the augmented variable given the diffusion model prior. Suppose we have a pre-trained score network $s_\psi(\mathcal{X}_t, t)$, following the EDM framework described in Section 2.2. The conditional distribution of $\mathcal{Z}$ is,

$$\mathcal{Z} \mid \cdots \sim \exp(-g(\mathcal{Z}) - \phi(\mathcal{Z}, \mathcal{X}; \rho)), \quad (12)$$

where $g(\mathcal{Z})$ is the implicit prior defined by the diffusion model. Since the distance $\phi$ is defined as the squared error in Eq. (5), sampling from the posterior Eq. (12) is essentially a

---

**Algorithm 1:** Gibbs sampler for DiffBCP

**Input:** Data $\mathcal{Y}$, Diffusion Model $s_\psi$, Coupling constant $c$, Hyper-priors.
**Output:** Posterior samples of CP factors $\boldsymbol{\lambda}, \mathbf{A}^{(1:N)}$.
**for** $i = 1, \ldots, M$ **do**
    Compute $\rho = \sqrt{c/\tau}$;
    Sample $\boldsymbol{\lambda}$ from Eq. (6);
    Sample $\mathbf{A}^{(n)}, \forall n$ from Eq. (7);
    adapt_factor($\boldsymbol{\lambda}, \mathbf{A}^{(1:N)}$);
    Sample $\tau$ from Eq. (8);
    Sample $\zeta_r, \forall r$ from Eq. (9);
    Sample $\nu_r, \forall r$ from Eq. (10);
    Sample $\theta_r, \forall r$ from Eq. (11);
    Find $T_i$ according to $\sigma(T_i) = \rho$;
    **for** $t = T_i, \ldots, 1$ **do**
        Denoising step by Eq. (2);
    **end**
    Collect samples after burn-in;
**end**

---

denoising problem with prior $g(\mathcal{Z})$ and observation $\mathcal{X}$ with noise level $\rho$ (Wu et al., 2024). Different from PnP-DM (Wu et al., 2024) where the guidance is unconstrained, here the guidance $\mathcal{X}$ is constrained by the CP decomposition structure. The sampling can be solved by running the EDM denoising process Eq. (2). Specifically, for given noise level $\rho$, we find the corresponding time step $t$ such that $\sigma(t) = \rho$. Then, we solve the SDE in Eq. (2) from $t$ to 0 to get the sample of $\mathcal{Z}$.

**Noise-adaptive coupling schedule**   As we will discuss in the next subsection, the coupling parameter $\rho$ plays an important role in the sampling process. Thus, we need to design a proper annealing schedule for $\rho$. In Wu et al. (2024); Xu & Chi (2024), heuristic and deterministic schedules are tested. However, the performances may be sensitive to the choice of scheduler. In our formulation, $\tau$ and $\rho$ always appear *jointly* in the conditional updates, *e.g.*, Eqs. (6) and (7). Thus the optimization/sampling landscape seen by the latent variables is shaped by the *relative* magnitude between $\tau$ and $\rho^{-2}$ rather than $\rho$ alone. Motivated by this, we set $\tau\rho^2 = c$, where $c$ is a hyperparameter constant. When $\tau$ is inferred during the sampling procedure, this rule automatically adjusts $\rho$ to maintain a fixed relative scale between the likelihood and coupling terms.

The overall algorithm is summarized in Algorithm 1. More details and derivation can be find in appendix Section A.

### 3.2.3. THEORETICAL PROPERTIES

In this subsection, we investigate the approximations in the sampling algorithm and how the noise parameter $\rho$ af-

fects the sampling process. Define the set of all hyperparameters $(\tau, \zeta, \nu, \theta)$ and CP factors $(\lambda, \mathbf{A}^{(1:N)})$ as $\Theta$. The $\rho$-smoothed prior induced by the diffusion model is,

$$p_\rho(\mathbfcal{X}) = \frac{1}{C_\rho} \int p(\mathbfcal{Z}) \exp(-\varphi(\mathbfcal{Z}, \mathbfcal{X}; \rho)) \, \mathrm{d}\mathbfcal{Z},$$

where $C_\rho$ is the normalizing constant. Denote the marginalized posterior as,

$$\pi_{\rho,\mathbfcal{X}}(\Theta) \propto p(\mathbfcal{Y} \mid \mathbfcal{X}(\Theta), \tau) p(\Theta) p_\rho(\mathbfcal{X}(\Theta)), \tag{13}$$

and the original posterior $\pi_0(\Theta)$ defined in Eq. (4). Under some conditions on the joint probability, Vono et al. (2019) shows that the smoothed posterior $\pi_{\rho,\mathbfcal{X}}(\Theta)$ converges to the original posterior $\pi_0(\Theta)$ as $\rho \to 0$, i.e.,

$$\lim_{\rho \to 0} D_{\mathrm{TV}}(\pi_{\rho,\mathbfcal{X}}, \pi_0) = 0. \tag{14}$$

This indicates that by choosing a smaller $\rho$, we can reduce the bias introduced by the split Gibbs sampler. However, using small $\rho$ also makes the denoising step more challenging, especially when the score function is approximated by a neural network. Let $\pi_\rho(\mathbfcal{Z}, \Theta)$ be the target joint density of the augmented form,

$$\pi_\rho(\mathbfcal{Z}, \Theta) \propto p(\mathbfcal{Y} \mid \mathbfcal{X}(\Theta), \tau) p(\Theta) p(\mathbfcal{Z}) \exp(-\phi(\mathbfcal{Z}, \mathbfcal{X}; \rho)), \tag{15}$$

with conditionals $\pi_\rho(\mathbfcal{Z} \mid \Theta)$ and $\pi_\rho(\Theta \mid \mathbfcal{Z})$, and let $\pi_{\rho,\mathbfcal{X}}(\Theta) = \int \pi_\rho(\mathbfcal{Z}, \Theta) \, d\mathbfcal{Z}$ be its marginal on $\Theta$. Assume that the practical diffusion sampler used in the denoising step induces a conditional density $q_\rho(\mathbfcal{Z} \mid \Theta) \approx \pi_\rho(\mathbfcal{Z} \mid \Theta)$ (for each fixed $\Theta$), and consider the inexact block Gibbs chain that alternates

$$\mathbfcal{Z}^{k+1} \sim q_\rho(\cdot \mid \Theta^k), \qquad \Theta^{k+1} \sim \pi_\rho(\cdot \mid \mathbfcal{Z}^{k+1}).$$

Assume moreover that the two families of conditionals $\pi_\rho(\Theta \mid \mathbfcal{Z})$ and $q_\rho(\mathbfcal{Z} \mid \Theta)$ are compatible, so that the above Markov chain admits a stationary joint distribution $\tilde{\pi}_\rho(\mathbfcal{Z}, \Theta)$. Denote $\tilde{\pi}_{\rho,\mathbfcal{X}}(\Theta) = \int \tilde{\pi}_\rho(\mathbfcal{Z}, \Theta) \, \mathrm{d}\mathbfcal{Z}$ and $\pi_{\rho,z}(\mathbfcal{Z}) = \int \pi_\rho(\mathbfcal{Z}, \Theta) \, \mathrm{d}\Theta$. The following theorem quantifies the stationary bias on $\Theta$ induced by the inexact diffusion denoising step.

**Theorem 3.4.** *For any $\Theta$ in the support of $\pi_{\rho,\mathbfcal{X}}$, the stationary bias on $\Theta$ can be quantified as*

$$D_{\mathrm{KL}}(\pi_{\rho,\mathbfcal{X}} \| \tilde{\pi}_{\rho,\mathbfcal{X}}) =$$
$$\mathbb{E}_{\Theta \sim \pi_{\rho,\mathbfcal{X}}} \left[ \log \mathbb{E}_{\mathbfcal{Z} \sim \pi_{\rho,z}} \left( \frac{q_\rho(\mathbfcal{Z} \mid \Theta)}{\pi_\rho(\mathbfcal{Z} \mid \Theta)} \right) \right]. \tag{16}$$

The proof follows Heurtel-Depeiges et al. (2024) and we defer it to appendix Section B.2. Eq. (14) and Theorem 3.4 can be used to explain the overall bias in the following corollary.

**Corollary 3.5.** *For any bounded measurable function $f$,*

$$\left| \mathbb{E}_{\tilde{\pi}_{\rho,\mathbfcal{X}}}[f(\Theta)] - \mathbb{E}_{\pi_{0,\mathbfcal{X}}}[f(\Theta)] \right| \leq$$
$$\left| \mathbb{E}_{\pi_{\rho,\mathbfcal{X}}}[f(\Theta)] - \mathbb{E}_{\pi_{0,\mathbfcal{X}}}[f(\Theta)] \right| +$$
$$\|f\|_\infty \sqrt{2 D_{\mathrm{KL}}(\pi_{\rho,\mathbfcal{X}} \| \tilde{\pi}_{\rho,\mathbfcal{X}})}, \tag{17}$$

*where the last term is given by Eq. (16).*

*Remark* 3.6. The first term on the RHS of Eq. (17) converges to 0 as $\rho \to 0$ by Eq. (14). The second term instead quantifies the stationary bias induced by the inexact diffusion step, as characterized in Theorem 3.4. It becomes smaller as $q_\rho(\mathbfcal{Z} \mid \Theta)$ gets closer to $\pi_\rho(\mathbfcal{Z} \mid \Theta)$, which is expected to happen when $\rho$ increases, since the denoising becomes easier. Thus there is a trade-off in the choice of $\rho$.

## 4. Experiments

Experiments are conducted on single NVIDIA A100 GPU with 40GB VRAM unless otherwise specified. More details and results are provided in appendix Section D. The code is available at https://github.com/taozerui/DiffBCP.

### 4.1. Image Inpainting and Denoising

In this experiment, we evaluate our method on natural image inpainting and denoising tasks, with pre-trained diffusion models on corresponding datasets as priors.

**Datasets and pre-trained models**  We adopt FFHQ (Karras et al., 2019) and ImageNet (Deng et al., 2009). We use the pre-trained diffusion model for both datasets with checkpoints from Chung et al. (2023). For each dataset, 128 images from the test set are randomly selected for evaluation. We adopt three types of masks for inpainting: (1) Uniform(0.9/0.7) randomly masks 90%/70% pixels in a uniform manner; (2) Stripe randomly masks pixels in horizontal and vertical stripes; (3) Irregular masks pixels using lines with random lengths and angles. Finally, all images are scaled to range $[0, 1]$ and iid Gaussian noise with standard variance $\sigma = 0.05$ is added for denoising.

**Baselines**  We compare with Bayesian CP (BCP, Zhao et al., 2015), Bayesian Tensor Ring (BTR, Long et al., 2021), Hierarchical Low-Rank Tensor Factorization (HLRTF, Luo et al., 2022), TD with Deep network priors (DeepTensor, Saragadam et al., 2024), TD with a tensor Correlated Total Variation regularization (tCTV, Wang et al., 2023), and TD with Global-LOcal-Nonlocal priors (GLON, Zhao et al., 2022). BCP and BTR are two traditional Bayesian TD for CP and TR decompositions. HLRTF and DeepTensor are two deep learning-based TD methods. GLON is a TD method that leverages pre-trained neural networks as priors.

*Table 1.* Results for FFHQ and ImageNet datasets. PSNR↑, SSIM↑, and LPIPS↓ metrics are reported, while SSIM and LPIPS are multiplied by 100 for better readability. Purple rows indicate TDs with deep (neural network) structures. Green rows indicate TDs with smooth or pre-trained model priors.

| Method | Uniform(0.7) | | | Uniform(0.9) | | | Stripe | | | Irregular | | |
|---|---|---|---|---|---|---|---|---|---|---|---|---|
| | PSNR | SSIM | LPIPS | PSNR | SSIM | LPIPS | PSNR | SSIM | LPIPS | PSNR | SSIM | LPIPS |
| *FFHQ* | | | | | | | | | | | | |
| BCP | 26.28 | 64.53 | 51.56 | 21.61 | 44.56 | 61.95 | 9.26 | 25.28 | 65.68 | 22.64 | 62.52 | 47.30 |
| BTR | 27.32 | 68.22 | 45.16 | 24.22 | 58.58 | 56.23 | 19.63 | 50.35 | 53.22 | 24.91 | 66.74 | 43.98 |
| HLRTF | 28.16 | 77.92 | 40.94 | 25.12 | 66.55 | 52.08 | 25.21 | 75.15 | 40.20 | 25.84 | 77.42 | 37.27 |
| DeepTensor | 28.23 | 72.23 | 30.83 | 26.11 | 61.66 | 40.65 | 26.44 | 75.70 | 29.53 | 28.01 | 79.49 | 26.19 |
| tCTV | 26.40 | 59.18 | 38.19 | 25.18 | 59.94 | 48.16 | 24.32 | 58.61 | 43.59 | 24.97 | 53.98 | 36.03 |
| GLON | 13.02 | 27.18 | 62.89 | 7.47 | 17.65 | 36.83 | 21.38 | 53.53 | 47.84 | 22.69 | 53.21 | 40.89 |
| DiffBCP | **32.13** | **88.83** | **17.70** | **28.28** | **81.18** | **28.93** | **27.91** | **85.11** | **19.89** | **30.34** | **88.60** | **15.98** |
| *ImageNet* | | | | | | | | | | | | |
| BCP | 24.34 | 61.07 | 47.97 | 20.17 | 39.71 | 59.46 | 10.05 | 28.62 | 62.56 | 21.33 | 61.21 | 41.93 |
| BTR | 25.03 | 63.44 | 43.58 | 21.80 | 48.37 | 56.15 | 18.68 | 49.45 | 49.79 | 22.44 | 60.81 | 42.94 |
| HLRTF | 26.00 | 73.94 | 38.56 | 22.65 | 59.35 | 50.66 | 23.30 | 70.09 | 38.34 | 22.01 | 71.83 | 35.73 |
| DeepTensor | 26.03 | 68.81 | 32.60 | 23.74 | 56.46 | 42.85 | 23.87 | 68.64 | 32.60 | 25.16 | 73.10 | 29.30 |
| tCTV | 24.94 | 60.13 | 36.78 | 23.03 | 54.83 | 48.50 | 22.87 | 58.14 | 41.47 | 23.62 | 55.96 | 33.89 |
| GLON | 12.47 | 27.46 | 60.45 | 7.02 | 15.39 | 70.43 | 20.65 | 53.14 | 45.60 | 21.54 | 54.86 | 41.01 |
| DiffBCP | **28.95** | **84.53** | **21.27** | **25.03** | **71.49** | **35.58** | **25.07** | **77.58** | **24.83** | **27.02** | **83.09** | **19.66** |

However, GLON only applies for pre-trained denoising networks with fixed noise levels.

**Results** We evaluate reconstruction quality using three complementary metrics: peak signal-to-noise ratio (PSNR) measuring pixel-wise accuracy, structural similarity index (SSIM) capturing perceptual similarity, and learned perceptual image patch similarity (LPIPS, Zhang et al., 2018) quantifying semantic fidelity. Higher PSNR/SSIM and lower LPIPS indicate better performance.

The quantitative results in Table 1 demonstrate that Diff-BCP consistently outperforms all baselines across different datasets and missing patterns. Specifically, on FFHQ with Uniform(0.9) mask, DiffBCP achieves 28.28 dB PSNR, improving over the best baseline (DeepTensor) by 2.17 dB. The gains are even more pronounced on Irregular masks (+2.33 dB) where structured priors matter most. On ImageNet, DiffBCP maintains strong performance on all metrics, with average improvements of 1.82 dB in PSNR and 12.42 in SSIM over the best competing method.

The qualitative comparisons in Fig. 1 reveal that DiffBCP reconstructs sharper textures and more coherent structures. Traditional multi-linear methods (BCP and BTR) produce overly coarse results due to limited prior expressiveness, *e.g.*, we can easily identify the low-rank structures. Non-linear decompositions (HLRTF and DeepTensor) generate finer-grained details but suffer from artifacts. GLON exhibits severe instability, particularly at high missing ratios (90%+), often converging to trivial solutions (predicting all

missing pixels as zero); this explains its poor quantitative performance in Table 1. In contrast, DiffBCP leverages the powerful diffusion prior while maintaining stable inference through the split Gibbs sampler.

We ablate the diffusion prior and the CUSP shrinkage prior in Section D.3, and report sensitivity to the scheduling hyperparameters $c$ and $\rho_{\min}$ in Section D.5. The two ablations show that (i) the diffusion prior is the main source of gain on structured masks, and (ii) CUSP enables strong recovery even when the initial CP rank is severely under-specified, so the model is not sensitive to the choice of initial rank.

### 4.2. Out-of-Distribution and High-Resolution Images

A critical challenge for practical deployment is handling images from distributions different from the diffusion prior's training data, as well as varying resolutions. To assess generalization capability, we evaluate on out-of-distribution (OOD) high-resolution images.

We select three diverse $2048 \times 2048$ images: Marseille (urban landscape), Tokyo (nighttime cityscape), and Westerlund (astronomical image). See appendix Section D for their sources. These images are deliberately chosen to be OOD relative to the FFHQ/ImageNet training distributions. The images also test multi-scale reasoning since the diffusion prior was trained on $256 \times 256$ images. We compare with PuTT (Loeschcke et al., 2024), a recent coarse-to-fine method achieving state-of-the-art results on high-dimensional tensors. For DiffBCP, we apply the reshape

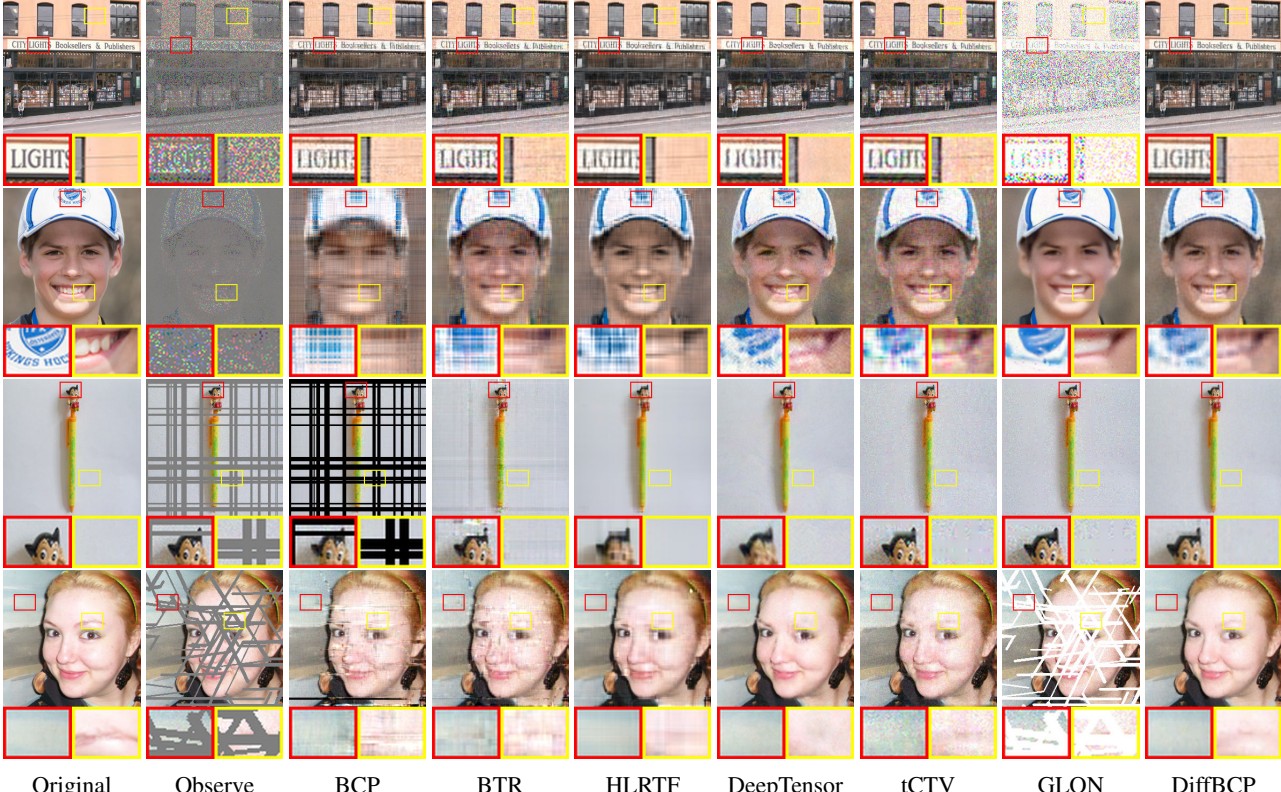

|  Original | Observe | BCP | BTR | HLRTF | DeepTensor | tCTV | GLON | DiffBCP |

*Figure 1.* Visualization for FFHQ and ImageNet datasets. From top to bottom: ImageNet with Uniform(0.7), FFHQ with Uniform(0.9), ImageNet with Stripe, and FFHQ with Irregular masks.

*Table 2.* Results for high-resolution images. PSNR↑ and SSIM↑ metrics are reported, while SSIM is multiplied by 100 for better readability.

| Method | Uni(0.9) | | Uni(0.95) | | Irr | |
|---|---|---|---|---|---|---|
| | PSNR | SSIM | PSNR | SSIM | PSNR | SSIM |
| *Marseille* | | | | | | |
| BCP | 16.94 | 29.42 | 15.89 | 26.55 | 17.63 | 33.55 |
| PuTT | 19.63 | 36.47 | 17.86 | 28.26 | 19.83 | 40.84 |
| DiffBCP | **20.15** | **39.08** | **18.01** | **28.42** | **20.94** | **46.75** |
| *Tokyo* | | | | | | |
| BCP | 17.89 | 37.61 | 16.90 | **36.67** | 18.62 | 40.40 |
| PuTT | 19.98 | 39.52 | 18.33 | 30.74 | 20.27 | 45.03 |
| DiffBCP | **20.66** | **44.90** | **18.90** | 35.50 | **21.36** | **51.38** |
| *Westerlund* | | | | | | |
| BCP | 21.12 | 65.60 | 20.08 | 65.09 | 22.26 | 68.34 |
| PuTT | **24.43** | 65.68 | **23.00** | 55.70 | 24.38 | 68.21 |
| DiffBCP | 23.85 | **68.33** | 21.86 | **66.16** | **25.27** | **74.03** |

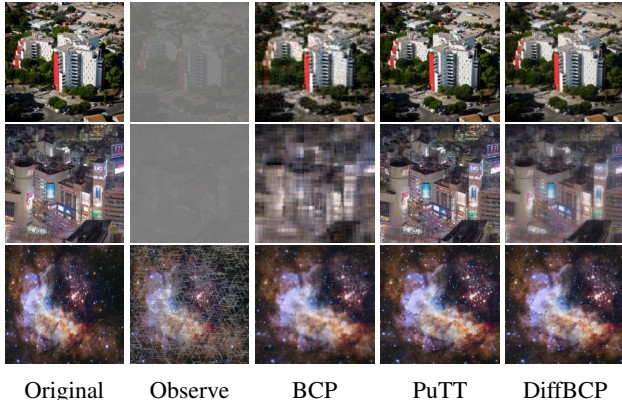

|  Original | Observe | BCP | PuTT | DiffBCP |

*Figure 2.* Visualization for high-resolution images. From top to bottom: Marseille with Uniform(0.9), Tokyo with Uniform(0.95), and Westerlund with Irregular masks.

operation in Eq. (5) to match the prior's expected input dimensions while preserving spatial structure.

Results in Table 2 show that DiffBCP maintains strong performance even on OOD high-resolution data. On Marseille with Irregular mask, DiffBCP achieves 20.94 dB PSNR ver-

sus 19.83 dB for PuTT, despite the significant distribution shift. Qualitative results (Fig. 2) reveal that DiffBCP better preserves fine details and global coherence. PuTT's coarse-to-fine strategy can introduce boundary artifacts when transitioning between scales, whereas our joint low-rank and diffusion modeling maintains consistency across resolutions. The relatively smaller gaps compared to in-distribution results suggest that the low-rank structure provides an inductive bias that partially compensates for distribution mismatch.

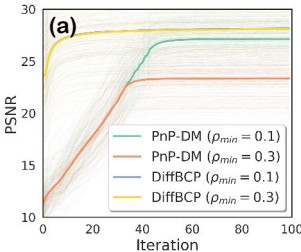
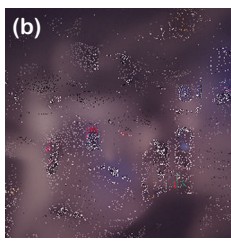

*Figure 3.* (a) Trace plot of PSNR for ImageNet with Uniform(0.7) mask. Thinner lines correspond to individual images and thicker lines show the average. (b) PnP-DM recovery of Tokyo with Uniform(0.95) mask.

### 4.3. Improving Diffusion Posterior Sampling

Finally, we investigate how the low-rank structure in Diff-BCP can improve diffusion posterior sampling, especially comparing with PnP-DM (Wu et al., 2024) which shares a similar split Gibbs sampler framework but without tensor decomposition. A direct quantitative comparison with DPS (Chung et al., 2023) and PnP-DM under both single-sample and posterior-mean evaluation is deferred to Section D.2.

**Faster mixing** As shown in Fig. 3(a), the low-rank assumption leads to an easier posterior distribution, which may lead to faster mixing (Roberts & Sahu, 2002). However, we should note that faster mixing does not necessarily lead to better performance, because the low-rank assumption may introduce bias if the data are not exactly low-rank.

**Robust to noise level and scheduling** Following Wu et al. (2024), we clip $\rho$ to $[\rho_{min}, \rho_{max}]$ in each sampling step. As shown in Fig. 3(a), PnP-DM could be sensitive to the scheduling of noise level (*e.g.*, $\rho_{min}$), while our noise-adaptive coupling schedule tends to be more robust. More results are shown in Section D.5.

**High-resolution images** As shown in Fig. 3(b), PnP-DM fails to recover the high-resolution images. As comparison, our method produces much better results as shown in Fig. 2. The corresponding wall-clock time and peak GPU memory are reported in Section D.7.

## 5. Conclusion and Discussion

We propose DiffBCP, a principled Bayesian framework for tensor decomposition that integrates pre-trained diffusion models as implicit priors. By combining the structural inductive bias of low-rank CP decomposition with the expressive power of diffusion priors, DiffBCP achieves state-of-the-art performance on tensor completion and denoising tasks. Key contributions include: (1) a cumulative shrinkage prior for automatic rank determination, (2) a split Gibbs sampler enabling tractable posterior inference despite the complex

coupling between likelihood, low-rank constraint, and diffusion prior, and (3) an adaptive noise scheduling strategy that removes hyperparameter sensitivity. The performance of DiffBCP rely on the underlying signal admitting a meaningful low-rank structure: when this assumption is violated, the structural contribution of the CP block vanishes while its factor updates remain comparatively more expensive than a pure diffusion sampler.

For future research, several directions merit further investigation. First, while we focus on tensor completion and denoising, the framework naturally extends to other linear inverse problems (e.g., compressed sensing, super-resolution). Second, the current implementation processes the full tensor in Steps Eqs. (6) and (7); stochastic mini-batch updates could reduce memory overhead for extremely large tensors. Third, we employ CP decomposition in this work, but the framework is compatible with other decompositions that may better capture specific structural patterns; we sketch the extension to tensor train and tensor ring in Section C. We hope this work inspires future research at the intersection of tensor decomposition and modern generative models.

## Acknowledgement

We thank the anonymous reviewers and AC for their constructive feedback. ZT was supported by the JSPS KAKENHI Grant Number JP26K21321 and RIKEN Incentive Research Project. QZ was supported by the JSPS KAKENHI Grant Number JP23K28109 and JSPS Bilateral Program Number JPJSBP120257420.

## Impact Statement

This paper presents work whose goal is to advance the field of machine learning, especially tensor decomposition and diffusion models. There are many potential societal consequences of our work, none of which we feel must be specifically highlighted here.

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

# A. Derivation of the Gibbs Sampler

The augmented joint distribution is,

$$\exp(-\ell(\mathcal{Y}; \mathcal{X}, \tau) - g(\mathcal{Z}) - \phi(\mathcal{Z}, \mathcal{X}; \rho) - f_\lambda(\boldsymbol{\lambda} \mid \boldsymbol{\theta}) - f_{\boldsymbol{\theta}}(\boldsymbol{\theta}) - f_{\mathbf{A}}(\mathbf{A}) - f_\tau(\tau)),$$

$$s.t. \quad \mathcal{X} = \mathsf{CP}\left(\boldsymbol{\lambda}, \mathbf{A}^{(1)}, \mathbf{A}^{(2)}, \dots, \mathbf{A}^{(N)}\right).$$

To enable Gibbs sampling of the spike-and-slab prior, we further introduce the augmented variable $\boldsymbol{\zeta} = (\zeta_1, \dots, \zeta_R)$, following $p(\zeta_r = l) = \omega_l, \forall l = 1, \dots, R$. This augmentation variable indicate which slab the component $r$ belongs to,

$$\theta_r \mid \zeta_r \sim \mathbb{1}(\zeta_r > r)\mathrm{InvGamma}(\alpha_\theta, \beta_\theta) + \mathbb{1}(\zeta_r \leq r)\delta_{\theta_\infty}, \qquad \forall r = 1, \dots, R,$$

where $\mathbb{1}(\cdot)$ is the indicator function. After the augmentation, the joint distribution becomes,

$$\exp(-\ell(\mathcal{Y}; \mathcal{X}, \tau) - g(\mathcal{Z}) - \phi(\mathcal{Z}, \mathcal{X}; \rho) - f_\lambda(\boldsymbol{\lambda} \mid \boldsymbol{\theta}) - f_{\boldsymbol{\theta}}(\boldsymbol{\theta} \mid \boldsymbol{\zeta}) - f_{\boldsymbol{\zeta}}(\boldsymbol{\zeta} \mid \boldsymbol{\nu}) - f_{\boldsymbol{\nu}}(\boldsymbol{\nu}) - f_{\mathbf{A}}(\mathbf{A}) - f_\tau(\tau)),$$

$$s.t. \quad \mathcal{X} = \mathsf{CP}\left(\boldsymbol{\lambda}, \mathbf{A}^{(1)}, \mathbf{A}^{(2)}, \dots, \mathbf{A}^{(N)}\right),$$

where the likelihood is defined as,

$$\ell(\mathcal{Y}; \mathcal{X}, \tau) = \frac{\tau}{2}\|\mathcal{O} * (\mathcal{Y} - \mathcal{X})\|_F^2,$$

and the priors are defined as,

$$\phi(\mathcal{Z}, \mathcal{X}; \rho) = \frac{1}{2\rho^2}\|\mathcal{Z} - \mathcal{X}\|_F^2,$$

$$f_\lambda(\boldsymbol{\lambda} \mid \boldsymbol{\theta}) = \frac{1}{2}\sum_{r=1}^{R}(\theta_r)^{-1}\lambda_r^2,$$

$$f_{\boldsymbol{\theta}}(\boldsymbol{\theta} \mid \boldsymbol{\zeta}) = \sum_{r=1}^{R}\mathrm{InvGamma}(\alpha_\theta, \beta_\theta)^{\mathbb{1}(\zeta_r > r)}\delta_{\theta_\infty}^{\mathbb{1}(\zeta_r \leq r)},$$

$$f_{\boldsymbol{\zeta}}(\boldsymbol{\zeta} \mid \boldsymbol{\nu}) = \mathrm{Categorical}(\boldsymbol{\omega}),$$

$$f_{\boldsymbol{\nu}}(\boldsymbol{\nu}) = \prod_{r=1}^{R}\mathrm{Beta}(\nu_r; 1, \beta),$$

$$f_{\mathbf{A}}(\mathbf{A}) = \frac{1}{2}\sum_{n=1}^{N}\|\mathbf{A}^{(n)}\|_F^2,$$

$$f_\tau(\tau) = (1 - \alpha_0)\log\tau + \kappa_0\tau.$$

## A.1. Latent Variable Steps

**Sample $\tau$** When sampling $\tau$, it is not dependent on $\phi$, thus we have,

$$\tau \mid \cdots \propto \exp(-f(\mathcal{Y}; \mathcal{X}, \tau) - l(\tau))$$

$$\propto \exp\left(-\frac{\tau}{2}\sum_{\mathbf{i} \in \Omega}(y_{\mathbf{i}} - x_{\mathbf{i}})^2\right) \cdot \exp(-(1 - \alpha_0)\log\tau - \kappa_0\tau)$$

$$\propto \exp\left(-\left(\kappa_0 + \frac{1}{2}\sum_{\mathbf{i} \in \Omega}(y_{\mathbf{i}} - x_{\mathbf{i}})^2\right)\tau\right) \cdot \exp\left(-(1 - \alpha_0 + \frac{|\Omega|}{2})\log\tau\right).$$

This is a Gamma distribution,

$$\tau \mid \cdots \sim \mathrm{Gamma}\left(\alpha_0 + \frac{|\Omega|}{2}, \kappa_0 + \frac{1}{2}\sum_{\mathbf{i} \in \Omega}(y_{\mathbf{i}} - x_{\mathbf{i}})^2\right).$$

**Sample A** The factors $\mathbf{A}$ are dependent on $f, h, \phi$. For CP decomposition, we have,

$$\mathbf{X}_{[n]} = \mathbf{A}^{(n)}\mathbf{\Lambda}\left(\mathbf{A}^{(N)} \odot \cdots \odot \mathbf{A}^{(n+1)} \odot \mathbf{A}^{(n-1)} \odot \cdots \odot \mathbf{A}^{(1)}\right)^{\mathsf{T}},$$

where $\mathbf{\Lambda} = \mathrm{diag}(\boldsymbol{\lambda})$ and $\odot$ is the Khatri-Rao product. For simplicity, we denote,

$$\mathbf{B}^{(n)} = \mathbf{\Lambda}\left(\mathbf{A}^{(N)} \odot \cdots \odot \mathbf{A}^{(n+1)} \odot \mathbf{A}^{(n-1)} \odot \cdots \odot \mathbf{A}^{(1)}\right)^{\mathsf{T}}.$$

The joint distribution becomes,

$$p(\mathbf{A}^{(n)} \mid \cdots) \propto \exp\left(-\frac{\tau}{2}\|\mathcal{O} * (\mathcal{Y} - \mathcal{X})\|_F^2 - \frac{1}{2}\|\mathbf{A}^{(n)}\|_F^2 - \frac{1}{2\rho^2}\|\mathcal{Z} - \mathcal{X}\|_F^2\right)$$

$$\propto \exp\left(-\frac{\tau}{2}\|\mathbf{O}_{[n]} * (\mathbf{Y}_{[n]} - \mathbf{A}^{(n)}\mathbf{B}^{(n)})\|_F^2 - \frac{1}{2}\|\mathbf{A}^{(n)}\|_F^2 - \frac{1}{2\rho^2}\|\mathbf{Z}_{[n]} - \mathbf{A}^{(n)}\mathbf{B}^{(n)}\|_F^2\right).$$

For $i = 1, \ldots, I_N$, we have

$$p(\mathbf{a}_{i:}^{(n)} \mid \cdots) \propto \exp\left(-\frac{\tau}{2}\|\mathbf{o}_{[n],i:} * (\mathbf{y}_{[n],i:} - \mathbf{a}_{i:}^{(n)}\mathbf{B}^{(n)})\|^2 - \frac{1}{2}\|\mathbf{a}_{i:}^{(n)}\|^2 - \frac{1}{2\rho^2}\|\mathbf{z}_{[n],i:} - \mathbf{a}_{i:}^{(n)}\mathbf{B}^{(n)}\|^2\right)$$

$$\propto \exp\left(-\frac{1}{2}\mathbf{a}_{i:}^{(n)}[\mathbf{I} + \tau(\mathbf{B}^{(n)}\mathrm{diag}(\mathbf{o}_{[n],i:})\mathbf{B}^{(n)\mathsf{T}}) + \frac{1}{\rho^2}(\mathbf{B}^{(n)}\mathbf{B}^{(n)\mathsf{T}})]\mathbf{a}_{i:}^{(n)\mathsf{T}} + \right.$$

$$\left. \mathbf{a}_{i:}^{(n)}(\tau\mathbf{y}_{[n],i:}\mathrm{diag}(\mathbf{o}_{[n],i:})\mathbf{B}^{(n)\mathsf{T}} + \frac{1}{\rho^2}\mathbf{z}_{[n],i:}\mathbf{B}^{(n)\mathsf{T}})\right).$$

This is a Gaussian distribution,

$$\mathbf{a}_i^{(n)} \mid \cdots \sim \mathcal{N}(\boldsymbol{\mu}_i^{(n)}, \boldsymbol{\Sigma}^{(n)}),$$

where,

$$\boldsymbol{\Sigma}^{(n)} = \left(\mathbf{I} + \tau(\mathbf{B}^{(n)}\mathrm{diag}(\mathbf{o}_{[n],i:})\mathbf{B}^{(n)\mathsf{T}}) + \frac{1}{\rho^2}(\mathbf{B}^{(n)}\mathbf{B}^{(n)\mathsf{T}})\right)^{-1},$$

$$\boldsymbol{\mu}_i^{(n)} = \boldsymbol{\Sigma}^{(n)}\left(\tau\mathbf{y}_{[n],i:}\mathrm{diag}(\mathbf{o}_{[n],i:})\mathbf{B}^{(n)\mathsf{T}} + \frac{1}{\rho^2}\mathbf{z}_{[n],i:}\mathbf{B}^{(n)\mathsf{T}}\right).$$

We can easily sample from this Gaussian distribution.

**Sample $\boldsymbol{\lambda}$** We express the CP decomposition as,

$$\mathcal{X} = \sum_{r=1}^{R} \lambda_r \mathbf{a}_{:r}^{(1)} \circ \mathbf{a}_{:r}^{(2)} \cdots \circ \mathbf{a}_{:r}^{(N)}$$

$$= \lambda_r \mathbf{a}_{:r}^{(1)} \circ \mathbf{a}_{:r}^{(2)} \cdots \circ \mathbf{a}_{:r}^{(N)} + \sum_{h=1,h\neq r}^{R} \lambda_h \mathbf{a}_{:h}^{(1)} \circ \mathbf{a}_{:h}^{(2)} \cdots \circ \mathbf{a}_{:h}^{(N)},$$

$$= \lambda_r \mathcal{C}^r + \mathcal{D}^r,$$

where we denote $\mathcal{C}^r = \mathbf{a}_{:r}^{(1)} \circ \mathbf{a}_{:r}^{(2)} \cdots \circ \mathbf{a}_{:r}^{(N)}$ and $\mathcal{D}^r = \sum_{h=1,h\neq r}^{R} \lambda_h \mathcal{C}^h$. For each component $r$, we have,

$$p(\lambda_r \mid \cdots) \propto \exp\left(-\frac{1}{2}\theta_r\lambda_r^2 - \frac{\tau}{2}\|\mathcal{O} * (\mathcal{Y} - \lambda_r\mathcal{C}^r - \sum_{r=1,h\neq r}\lambda_h\mathcal{C}^h)\|_F^2 - \frac{1}{2\rho^2}\|\mathcal{Z} - \lambda_r\mathcal{C}^r - \sum_{r=1,h\neq r}\lambda_h\mathcal{C}^h\|_F^2\right)$$

$$\propto \exp\left(-\frac{1}{2}\left[\theta_r + \tau\sum_{\mathbf{i}\in\Omega}c_{\mathbf{i}}^2 + \frac{1}{\rho^2}\sum_{\mathbf{i}}c_{\mathbf{i}}^2\right]\lambda_r^2 + \left[\tau\sum_{\mathbf{i}\in\Omega}c_{\mathbf{i}}(y_{\mathbf{i}} - d_{\mathbf{i}}) + \frac{1}{\rho^2}\sum_{\mathbf{i}}c_{\mathbf{i}}(z_{\mathbf{i}} - d_{\mathbf{i}})\right]\lambda_r\right).$$

This is a Gaussian distribution,

$$\lambda_r | \cdots \sim \mathcal{N}(\mu_r, \sigma_r),$$

where,

$$\sigma_r = \left( \theta_r + \tau \| \mathbf{O} * \mathbf{C}^r \|_F^2 + \frac{1}{\rho^2} \| \mathbf{C}^r \|_F^2 \right)^{-1},$$

$$\mu_r = \sigma_r \mathsf{Sum} \left( \tau \mathbf{O} * \mathbf{C}^r * (\mathbf{Y} - \mathbf{D}^r) + \frac{1}{\rho^2} \mathbf{C}^r * (\mathbf{Z} - \mathbf{D}^r) \right),$$

with $\mathsf{Sum}(\cdot)$ the sum of all elements.

**Sample $\theta$**  For each component $r$, the full conditional distribution of $\theta_r$ is,

$$p(\theta_r | \cdots) \propto \exp(-\frac{1}{2}\theta_r \lambda_r^2) \cdot \mathrm{InvGamma}(\theta_r; \alpha_\theta, \beta_\theta)^{\mathbb{1}(\zeta_r > r)} \cdot \delta_{\theta_\infty}^{\mathbb{1}(\zeta_r \le r)}$$

$$\propto \theta_r^{-(\alpha_\theta + 1)} \exp\left( - \left( \beta_\theta + \frac{\lambda_r^2}{2} \right) \theta_r^{-1} \right)^{\mathbb{1}(\zeta_r > r)} \cdot \delta_{\theta_\infty}^{\mathbb{1}(\zeta_r \le r)}.$$

Thus, we have,

$$\theta_r | \cdots \sim \mathrm{InvGamma}\left( \alpha_\theta + \frac{1}{2}, \beta_\theta + \frac{\lambda_r^2}{2} \right)^{\mathbb{1}(\zeta_r > r)} \cdot \delta_{\theta_\infty}^{\mathbb{1}(\zeta_r \le r)}.$$

**Sample $\zeta$**  To sample $\zeta_r$, we can marginalize out $\theta_r$. If $\zeta_r \le r$, we have,

$$p(\zeta_r = l | \cdots) \propto \int p(\zeta_r = l) \cdot p(\theta_r | \zeta_r) \cdot p(\lambda_r | \theta_r) \, \mathrm{d}\theta_r$$

$$\propto \int \omega_l \delta_{\theta_\infty} \mathrm{Normal}(\lambda_r; 0, \theta_r) \, \mathrm{d}\theta_r$$

$$= \omega_l \mathrm{Normal}(\lambda_r; 0, \theta_\infty).$$

Else, if $\zeta_r > r$, we have,

$$p(\zeta_r = l | \cdots) \propto \int p(\zeta_r = l) \cdot p(\theta_r | \zeta_r) \cdot p(\lambda_r | \theta_r) \, \mathrm{d}\theta_r$$

$$\propto \int \omega_l \mathrm{InvGamma}(\theta_r; \alpha_\theta, \beta_\theta) \cdot \mathrm{Normal}(\lambda_r; 0, \theta_r) \, \mathrm{d}\theta_r$$

$$= \omega_l \cdot \mathrm{Student\text{-}t}_{2\alpha_\theta}(\lambda_r; 0, \beta_\theta / \alpha_\theta).$$

**Sample $\nu$**  From $r = 1$ to $R - 1$, we have,

$$p(\nu_r | \cdots) \propto p(\nu_r) \cdot p(\zeta | \nu_r)$$

$$\propto \mathrm{Beta}(\nu_r; 1, \beta) \cdot \prod_{h=r}^{R} \omega_{\zeta_h}$$

$$\propto \nu_r^{\sum_{h=r}^{R} \mathbb{1}(\zeta_h = r)} (1 - \nu_r)^{\beta - 1 + \sum_{h=r+1}^{R} \mathbb{1}(\zeta_h > r)}.$$

This is a Beta distribution,

$$\nu_r | \cdots \sim \mathrm{Beta}\left( 1 + \sum_{h=1}^{R} \mathbb{1}(\zeta_h = r), \beta + \sum_{h=1}^{R} \mathbb{1}(\zeta_h > r) \right).$$

**Rank adaptation**  We follow Rai et al. (2014); Bhattacharya & Dunson (2011) to adaptively adjust the number of components in $\boldsymbol{\lambda}$ (the CP rank). At time step $i$, we adapt the rank with probability $p(i) = \exp(\alpha_0 + \alpha_1 i)$, where $\alpha_0$ and $\alpha_1$ are hyperparameters. If there are inactive factors, we remove them. If all factors are active, we add a new factor (and associated parameters) from the priors. Details are shown in Algorithm 2.

---

**Algorithm 2:** Rank adaptation adapt_factor($\boldsymbol{\lambda}, \mathbf{A}$)

---

**Input:** CP factors $\boldsymbol{\lambda}, \mathbf{A}_1, \ldots, \mathbf{A}_D$, hyperparameters $\alpha_0, \alpha_1$, current iteration $i$.

**if** rand() $\leq \exp(\alpha_0 + \alpha_1 i)$ **then**

    // *Adapt with probability* $\exp(\alpha_0 + \alpha_1 i)$

    **if** $\min(\|\lambda_r\|) < 10^{-4}$ **then**

        Remove all components $r$ with $|\lambda_r| < 10^{-4}$;

    **else**

        Add a new component with $\lambda_{\text{new}} \sim \mathcal{N}(0, 1)$ and $\mathbf{a}_{\text{new}}^{(d)} \sim \mathcal{N}(\mathbf{0}, \mathbf{I}), \forall d = 1, \ldots, D$;

    **end**

**end**

---

## B. Proof of Theorems

### B.1. Proof of Theorem 3.1

*Proof.* In the BCP with CUSP prior, since $\lambda_r$ and $a_{i_n r}^{(n)}$ are independent, and the means are zero, we have,

$$\mathbb{E}[\lambda_r \prod_{n=1}^{N} a_{i_n r}^{(n)}]^2 = \mathbb{E}\lambda_r^2 \prod_{n=1}^{N} \mathbb{E}[a_{i_n r}^{(n)}]^2$$

$$\leq \mathbb{E}\lambda_r^2 \prod_{n=1}^{N} \eta_n, \tag{18}$$

where $\eta_n$ bounds $\mathbb{E}[a_{i_n r}^{(n)}]^2$. For the first term, we have

$$\mathbb{E}\lambda_r^2 = \mathbb{E}[\mathbb{E}\lambda_r^2 \mid \theta_r]$$

$$= \mathbb{E}[1 - \pi_r] \frac{\beta_\theta}{\alpha_\theta - 1} + \mathbb{E}[\pi_r] \theta_\infty.$$

For the expectation of $\pi_r$, we have,

$$\mathbb{E}[\pi_r] = \sum_{l=1}^{r} \mathbb{E}[\omega_l]$$

$$= \sum_{l=1}^{r} \mathbb{E}[\nu_l] \prod_{m=1}^{l-1} (1 - \mathbb{E}[\nu_m])$$

$$= \sum_{l=1}^{r} \frac{1}{1 + \beta} \left( \frac{\beta}{1 + \beta} \right)^{l-1}$$

$$= 1 - \left( \frac{\beta}{1 + \beta} \right)^r.$$

Thus, we have,

$$\mathbb{E}\lambda_r^2 = \left( \frac{\beta}{1 + \beta} \right)^r \left( \frac{\beta_\theta}{\alpha_\theta - 1} - \theta_\infty \right) + \theta_\infty. \tag{19}$$

Injecting Eq. (19) into Eq. (18), we get,

$$\mathbb{E}[\lambda_r \prod_{n=1}^{N} a_{i_n r}^{(n)}]^2 \leq \left[ \left( \frac{\beta}{1 + \beta} \right)^r \left( \frac{\beta_\theta}{\alpha_\theta - 1} - \theta_\infty \right) + \theta_\infty \right] \prod_{n=1}^{N} \eta_n.$$

Using the Markov inequality, we have,

$$p((x_{r,\mathbf{i}})^2 \geq \epsilon) \leq \frac{\mathbb{E}[(x_{r,\mathbf{i}})^2]}{\epsilon}$$

$$\leq \frac{1}{\epsilon} \left[ \left(\frac{\beta}{1+\beta}\right)^r \left(\frac{\beta_\theta}{\alpha_\theta - 1} - \theta_\infty\right) + \theta_\infty \right] \prod_{n=1}^N \eta_n,$$

which concludes the proof. □

### B.2. Proof of Theorem 3.4

*Proof.* First, by Bayes' rule, for any $(\mathcal{Z}, \Theta)$ with $\pi_\rho(\Theta \mid \mathcal{Z}) > 0$,

$$\pi_{\rho,\mathcal{Z}}(\mathcal{Z}) = \pi_{\rho,\mathcal{X}}(\Theta) \frac{\pi_\rho(\mathcal{Z} \mid \Theta)}{\pi_\rho(\Theta \mid \mathcal{Z})}.$$

Integrating both sides over $\mathcal{Z}$ yields $1 = \pi_{\rho,\mathcal{X}}(\Theta) \int \frac{\pi_\rho(\mathcal{Z}|\Theta)}{\pi_\rho(\Theta|\mathcal{Z})} d\mathcal{Z}$, hence

$$\pi_{\rho,\mathcal{X}}(\Theta) = \left( \int \frac{\pi_\rho(\mathcal{Z} \mid \Theta)}{\pi_\rho(\Theta \mid \mathcal{Z})} d\mathcal{Z} \right)^{-1}. \tag{20}$$

Similarly, using the compatibility assumption, $\tilde{\pi}_\rho(\Theta \mid \mathcal{Z}) = \pi_\rho(\Theta \mid \mathcal{Z})$ and $\tilde{\pi}_\rho(\mathcal{Z} \mid \Theta) = q_\rho(\mathcal{Z} \mid \Theta)$ imply

$$\tilde{\pi}_{\rho,\mathcal{Z}}(\mathcal{Z}) = \tilde{\pi}_{\rho,\mathcal{X}}(\Theta) \frac{q_\rho(\mathcal{Z} \mid \Theta)}{\pi_\rho(\Theta \mid \mathcal{Z})}.$$

Integrating over $\mathcal{Z}$ yields

$$\tilde{\pi}_{\rho,\mathcal{X}}(\Theta) = \left( \int \frac{q_\rho(\mathcal{Z} \mid \Theta)}{\pi_\rho(\Theta \mid \mathcal{Z})} d\mathcal{Z} \right)^{-1}. \tag{21}$$

Subtracting (20) and (21) in reciprocal form gives

$$\frac{\tilde{\pi}_{\rho,\mathcal{X}}(\Theta) - \pi_{\rho,\mathcal{X}}(\Theta)}{\pi_{\rho,\mathcal{X}}(\Theta)\tilde{\pi}_{\rho,\mathcal{X}}(\Theta)} = \int \frac{\pi_\rho(\mathcal{Z} \mid \Theta) - q_\rho(\mathcal{Z} \mid \Theta)}{\pi_\rho(\Theta \mid \mathcal{Z})} d\mathcal{Z}.$$

Using again $\pi_{\rho,\mathcal{Z}}(\mathcal{Z}) = \pi_{\rho,\mathcal{X}}(\Theta)\frac{\pi_\rho(\mathcal{Z}|\Theta)}{\pi_\rho(\Theta|\mathcal{Z})}$, we obtain

$$\frac{\tilde{\pi}_{\rho,\mathcal{X}}(\Theta) - \pi_{\rho,\mathcal{X}}(\Theta)}{\tilde{\pi}_{\rho,\mathcal{X}}(\Theta)} = 1 - \int \frac{q_\rho(\mathcal{Z} \mid \Theta)}{\pi_\rho(\mathcal{Z} \mid \Theta)} \pi_{\rho,\mathcal{Z}}(\mathcal{Z}) d\mathcal{Z},$$

which rearranges,

$$\frac{\pi_{\rho,\mathcal{X}}(\Theta)}{\tilde{\pi}_{\rho,\mathcal{X}}(\Theta)} = \int \frac{q_\rho(\mathcal{Z} \mid \Theta)}{\pi_\rho(\mathcal{Z} \mid \Theta)} \pi_{\rho,\mathcal{Z}}(\mathcal{Z}) d\mathcal{Z}.$$

Taking logarithm and expectation w.r.t. $\Theta \sim \pi_{\rho,\mathcal{X}}$ yields (16). Pinsker's inequality concludes the TV bound. □

### B.3. Proof of Corollary 3.5

*Proof.* By the triangle inequality,

$$\left| \mathbb{E}_{\tilde{\pi}_{\rho,\mathcal{X}}}[f] - \mathbb{E}_{\pi_{0,\mathcal{X}}}[f] \right| \leq \left| \mathbb{E}_{\tilde{\pi}_{\rho,\mathcal{X}}}[f] - \mathbb{E}_{\pi_{\rho,\mathcal{X}}}[f] \right| + \left| \mathbb{E}_{\pi_{\rho,\mathcal{X}}}[f] - \mathbb{E}_{\pi_{0,\mathcal{X}}}[f] \right|.$$

For the first term, by the characterization of the total variation distance $|\mathbb{E}_P[f] - \mathbb{E}_Q[f]| \leq 2\|f\|_\infty D_{\mathrm{TV}}(P \| Q)$ for bounded measurable $f$, we have,

$$\left| \mathbb{E}_{\tilde{\pi}_{\rho,\mathcal{X}}}[f] - \mathbb{E}_{\pi_{\rho,\mathcal{X}}}[f] \right| \leq 2\|f\|_\infty D_{\mathrm{TV}}(\tilde{\pi}_{\rho,\mathcal{X}} \| \pi_{\rho,\mathcal{X}}).$$

Using the Pinsker's inequality $D_{\mathrm{TV}}(P \| Q) \leq \sqrt{\frac{1}{2}\mathrm{KL}(P \| Q)}$, we have,

$$\left| \mathbb{E}_{\tilde{\pi}_{\rho,\mathcal{X}}}[f] - \mathbb{E}_{\pi_{0,\mathcal{X}}}[f] \right| \leq \left| \mathbb{E}_{\pi_{\rho,\mathcal{X}}}[f] - \mathbb{E}_{\pi_{0,\mathcal{X}}}[f] \right| + 2\|f\|_\infty D_{\mathrm{TV}}(\tilde{\pi}_{\rho,\mathcal{X}} \| \pi_{\rho,\mathcal{X}})$$

$$\leq \left| \mathbb{E}_{\pi_{\rho,\mathcal{X}}}[f] - \mathbb{E}_{\pi_{0,\mathcal{X}}}[f] \right| + \|f\|_\infty \sqrt{2D_{\mathrm{KL}}(\tilde{\pi}_{\rho,\mathcal{X}} \| \pi_{\rho,\mathcal{X}})}.$$

□

## C. Generalization to Tensor Train and Tensor Ring Decompositions

The DiffBCP framework is largely agnostic to the specific low-rank decomposition. Replacing CP with other decompositions such as TT (Oseledets, 2011) or TR (Zhao et al., 2016) only requires changing the contraction map $\mathcal{X}(\Theta)$ that connects latent factors to the reconstructed tensor; the likelihood $p(\mathcal{Y} \mid \mathcal{X}, \tau)$, the augmented variable block on $\mathcal{Z}$, and the overall split Gibbs framework remain unchanged.

Concretely, for an $N$-th order tensor $\mathcal{X} \in \mathbb{R}^{I_1 \times \cdots \times I_N}$, a TT decomposition writes each entry as

$$x_{\underline{\mathbf{i}}} = \mathcal{G}_{i_1}^{(1)} \mathcal{G}_{i_2}^{(2)} \cdots \mathcal{G}_{i_N}^{(N)},$$

where $\mathcal{G}^{(n)} \in \mathbb{R}^{R_{n-1} \times I_n \times R_n}$ are the TT cores with $R_0 = R_N = 1$; TR relaxes the boundary to $R_0 = R_N$ and replaces the matrix product with a trace. A key observation is that, conditional on all other cores, each core $\mathcal{G}^{(n)}$ enters the reconstruction *linearly*. Placing an isotropic Gaussian prior on the entries of each core therefore preserves Gaussian conjugacy for its full conditional given $\mathcal{Y}, \mathcal{Z}, \tau$, and the other cores, yielding the same matrix-normal update form as the CP factor update in Eq. (7), with $\mathbf{B}^{(n)}$ replaced by the appropriate contraction of the remaining cores. The CUSP rank-shrinkage prior generalizes analogously: it can be placed on the slices of $\mathcal{G}^{(n)}$ along the rank dimensions $R_{n-1}, R_n$ to shrink redundant TT/TR ranks, and the Gibbs updates of the CUSP auxiliaries remain unchanged because they depend only on the prior structure, not on the specific form of $\mathcal{X}(\Theta)$.

The diffusion-prior block in Eq. (12) is independent of the choice of decomposition, since it only depends on $\mathcal{X}$ as a noisy observation of $\mathcal{Z}$ via the coupling $\phi(\mathcal{Z}, \mathcal{X}; \rho)$. Consequently, the theoretical results in Theorem 3.4 and Corollary 3.5 also hold verbatim, as they are stated in terms of the augmented posterior $\pi_\rho(\mathcal{Z}, \Theta)$ without any CP-specific structure. A full empirical study of DiffBCP for TT/TR decompositions is left to future work.

## D. Experiments

### D.1. Image Inpainting and Denoising

**Implementations** For baseline models, we use their official implementations. BCP, BTR, tCTV, and GLON are implemented in MATLAB. We run them on a server with Intel Xeon Silver 4316 CPU@2.30GHz CPU and 512 GB RAM. HLRTF, DeepTensor, and our method are implemented in PyTorch.

**Hyperparameters** Bayesian tensor decomposition including BCP and BTR uses fully Bayesian inference, so they do not require extensive hyperparameter tuning. We choose CP rank 100 and TR rank 30, which are sufficiently large for the $256 \times 256 \times 3$ images. For other baselines, we tune their hyperparameters for the best performance on the validation set.

For our method, we re-arrange images into patches of shape $16 \times 16 \times 3$ with strite 8, so the resulting image tensor has shape $16 \times 16 \times 3 \times 961$. We set the CP rank as 200 and adaptively tune the rank during sampling. The hyperparameters for the CUSP prior are set as follows,

$$\alpha_0 = 1\mathrm{e}{-3}, \ \kappa_0 = 1\mathrm{e}{-3}, \ \beta = 5.0, \ \alpha_\theta = 2.0, \ \beta_\theta = 2.0, \ \theta_\infty = 1\mathrm{e}{-3}.$$

Most of these hyperparameters are noninformative and we do not tune them. The constant for the coupling parameter is set as $c = 100.0$, and we further clip $\rho$ into $[0.3, 10.0]$ following Wu et al. (2024). We study the sensitivity to both $c$ and $\rho_{\min}$ in Section D.5. For the Gibbs sampler, we run 100 iterations in total with 40 burn-in iterations. We collect one sample every four iterations after burn-in.

**Evaluation** We report averaged value of peak signal-to-noise ratio (PSNR), structural similarity index (SSIM), and learned perceptual image patch similarity (LPIPS). PSNR and SSIM metrics are computed using the implementation from the `MONAI` library (Cardoso et al., 2022) with default parameters. LPIPS is computed using the implementation in the VQGAN project[1].

### D.2. Comparison with Diffusion Posterior Samplers

We further compare DiffBCP against representative diffusion posterior samplers, including DPS (Chung et al., 2023) and PnP-DM (Wu et al., 2024), on the FFHQ inpainting benchmark. We use the same pre-trained diffusion checkpoint and the

---

[1] https://github.com/CompVis/taming-transformers

*Table 3.* Comparison with diffusion posterior samplers (DPS and PnP-DM) on FFHQ inpainting. PSNR↑, SSIM↑, and LPIPS↓ are reported, with SSIM and LPIPS multiplied by 100 for readability. "Single-sample reconstruction" denotes the reconstruction from a single posterior sample; "Posterior-mean reconstruction" denotes the posterior-mean estimate averaged over the collected samples. DPS does not produce multiple posterior samples and is therefore evaluated in the single-sample regime only. Within each block, bold marks the best value per column.

| | Uniform(0.7) | | | Uniform(0.9) | | | Stripe | | | Irregular | | |
|---|---|---|---|---|---|---|---|---|---|---|---|---|
| Method | PSNR | SSIM | LPIPS | PSNR | SSIM | LPIPS | PSNR | SSIM | LPIPS | PSNR | SSIM | LPIPS |
| *Single-sample reconstruction* | | | | | | | | | | | | |
| DPS (Chung et al., 2023) | 30.75 | **87.31** | **15.61** | 26.98 | **78.81** | **21.22** | 27.37 | **85.27** | **14.84** | **29.87** | **88.70** | **12.67** |
| PnP-DM (Wu et al., 2024) | 29.45 | 70.87 | 24.65 | 27.21 | 65.30 | 31.91 | 25.82 | 68.20 | 25.42 | 28.04 | 71.04 | 22.79 |
| DiffBCP | **31.49** | 85.94 | 21.31 | **27.80** | 77.24 | 33.67 | **27.51** | 81.36 | 23.93 | 29.84 | 85.53 | 19.71 |
| *Posterior-mean reconstruction* | | | | | | | | | | | | |
| PnP-DM (Wu et al., 2024) | **33.20** | **88.92** | **17.30** | **28.41** | **81.22** | **25.88** | 27.30 | **86.10** | **17.74** | **30.56** | **89.38** | **14.60** |
| DiffBCP | 32.13 | 88.83 | 17.70 | 28.28 | 81.18 | 28.93 | **27.91** | 85.11 | 19.89 | 30.34 | 88.60 | 15.98 |

*Table 4.* Ablation study on FFHQ inpainting. "Full" is our default DiffBCP with both the CUSP shrinkage prior and the diffusion-prior block, using an initial CP rank of $R = 200$. "w/o DM" removes the diffusion-prior block while keeping CUSP. "rank=$R$, w/ CUSP" keeps CUSP active but starts from an initial CP rank of $R$. "rank=$R$" disables CUSP and fixes the CP rank at $R$. The Full row (highlighted in green) achieves the best value in every column; the other rows are ordered to isolate the contribution of each component.

| | Uniform(0.7) | | | Uniform(0.9) | | | Stripe | | | Irregular | | |
|---|---|---|---|---|---|---|---|---|---|---|---|---|
| Method | PSNR | SSIM | LPIPS | PSNR | SSIM | LPIPS | PSNR | SSIM | LPIPS | PSNR | SSIM | LPIPS |
| DiffBCP (Full) | **32.13** | **88.83** | **17.70** | **28.28** | **81.18** | **28.93** | **27.91** | **85.11** | **19.89** | **30.34** | **88.60** | **15.98** |
| *Ablating the diffusion-prior block* | | | | | | | | | | | | |
| DiffBCP (w/o DM) | 32.05 | 88.00 | 18.88 | 28.10 | 79.24 | 32.52 | 25.82 | 79.35 | 28.97 | 28.64 | 85.28 | 21.39 |
| *CUSP under rank misspecification (adaptive vs. fixed rank)* | | | | | | | | | | | | |
| DiffBCP (rank=10, w/ CUSP) | 29.60 | 84.69 | 23.86 | 27.64 | 79.40 | 31.41 | 26.51 | 79.38 | 28.51 | 28.12 | 82.62 | 25.58 |
| DiffBCP (rank=10) | 26.35 | 75.17 | 37.39 | 25.83 | 73.42 | 40.75 | 23.60 | 68.80 | 44.17 | 25.23 | 72.68 | 40.64 |
| DiffBCP (rank=100) | 31.79 | 87.96 | 18.99 | 28.09 | 79.20 | 32.28 | 25.56 | 78.42 | 29.76 | 28.33 | 84.44 | 22.36 |
| DiffBCP (rank=200) | 32.09 | 87.85 | 19.06 | 28.06 | 78.80 | 33.07 | 25.84 | 79.55 | 28.92 | 28.75 | 85.52 | 21.30 |
| DiffBCP (rank=300) | 32.11 | 87.58 | 19.33 | 28.01 | 78.62 | 33.46 | 25.95 | 79.89 | 28.83 | 28.92 | 85.80 | 21.10 |

same masks as in Section D.1, so that all three methods share an identical observation model and prior network.

For PnP-DM and DiffBCP, both of which produce posterior samples via a split Gibbs framework, we report two evaluation modes. The *single-sample* reconstruction uses a single posterior sample as the reconstruction, which directly reflects the quality of a single draw from the posterior. The *posterior-mean* reconstruction averages the collected samples after burn-in (see Section D.1 for the sampling schedule), and is closer in spirit to a minimum-mean-squared-error estimator. DPS does not produce a posterior distribution, and is therefore evaluated in the single-sample regime only.

Table 3 summarizes the results. Under *single-sample* evaluation, DiffBCP attains the best PSNR on three of the four masks (Uniform 0.7/0.9 and Stripe) and is essentially tied with DPS on Irregular, while consistently improving over PnP-DM on PSNR and SSIM across all four masks. Compared with DPS, however, DiffBCP trails on SSIM and LPIPS in all four masks. We interpret this as a pixel-vs-perceptual trade-off: the explicit low-rank CP structure regularizes each individual posterior sample and brings it closer to the underlying signal in $\ell_2$ sense, but it also discourages the kind of high-frequency texture that perceptual metrics (SSIM and especially LPIPS) reward. DPS, which collapses to a single MAP-like reconstruction guided by the diffusion prior, is comparatively well aligned with this perceptual axis.

Under *posterior-mean* evaluation, DiffBCP is competitive with PnP-DM across all metrics, with the two methods alternating as the best across the twelve metric/mask cells.

### D.3. Ablation Studies

We ablate the two main components of DiffBCP — the diffusion-prior block on $\mathcal{Z}$ and the CUSP shrinkage prior on the CP factor weights — on the FFHQ inpainting benchmark from Section D.1. Table 4 reports posterior-mean reconstruction quality for the Full model, a variant without the diffusion-prior block, and a series of variants where CUSP is replaced by a fixed CP rank. For such case, we replace the CUSP prior with simple Gaussian priors on the CP factors.

**Contribution of the diffusion-prior block.** Comparing Full and "w/o DM" shows that the diffusion prior contributes only a small PSNR gain on uniformly random masks (e.g., $+0.08\,\text{dB}$ on Uniform 0.7, $+0.18\,\text{dB}$ on Uniform 0.9), but a substantially larger gain on structured corruption: $+2.09\,\text{dB}$ on Stripe and $+1.70\,\text{dB}$ on Irregular, with the LPIPS gap widening even more (e.g., 19.89 vs. 28.97 on Stripe). This pattern is consistent with the intuition that uniformly missing pixels are mostly recoverable from low-rank structure alone, whereas stripe and irregular masks remove contiguous spatial structure that the data-driven diffusion prior is uniquely positioned to fill in.

**CUSP under rank misspecification.** The remaining rows isolate the role of CUSP by removing the shrinkage prior and fixing the CP rank. The pure "rank=10" variant collapses across all masks, reflecting severe under-parameterization. Activating CUSP on top of the same initial rank of 10 ("rank=10, w/ CUSP") substantially mitigates this failure. A residual gap to Full remains, which we attribute to the limited Gibbs budget used in the ablation: CUSP introduces and activates new components incrementally, and at 100 Gibbs iterations a chain that starts at $R = 10$ has not yet reached its equilibrium rank — the corresponding trajectory in Fig. 4 is still expanding past iteration 100 and only plateaus around iteration 400. By contrast, the Full model starts at $R = 200$ (close to or above the equilibrium for these masks), so its CP factor pool only needs to be shrunk, which converges within the same 100-iteration budget. The practical takeaway is that CUSP makes the model *robust* to misspecified initial rank but does not eliminate it as a hyperparameter when the sampling budget is tight; a moderately large initial rank remains preferable, and an under-specified initial rank can be compensated by lengthening the chain. Finally, simply increasing the fixed rank to 100, 200, or 300 without CUSP closes the gap on uniform masks but does *not* reach Full on the structured masks, confirming that the contribution of CUSP is not reducible to simply allocating a larger CP rank.

### D.4. Effective Rank Evolution under CUSP

To complement the static ablation in Section D.3, we track how the effective CP rank evolves over the course of the Gibbs sampler when CUSP is enabled. For each setting we run 10 FFHQ images and record the active component count $|\{r : |\lambda_r| \geq 10^{-4}\}|$ at every iteration (cf. Algorithm 2). We extend the chain to 500 Gibbs iterations in this analysis because rank adaptation under CUSP converges noticeably more slowly than the reconstruction metrics.

Fig. 4 shows the characteristic behavior on the Uniform(0.9) mask. The chain initialized at $R = 10$ monotonically *expands* its effective rank — CUSP activates additional components in response to residual fitting error — while the chains initialized at $R \in \{100, 200, 300\}$ all *shrink* their effective rank. By 500 iterations the four trajectories collapse onto a narrow common band, with the spread reflecting genuine per-image variability in the underlying low-rank structure together with the slow tail of rank adaptation when starting far above equilibrium. This is a direct empirical confirmation that CUSP is *bidirectionally adaptive*: it prunes redundant components when over-specified and expands the active set when under-specified, removing the need to choose the initial CP rank carefully.

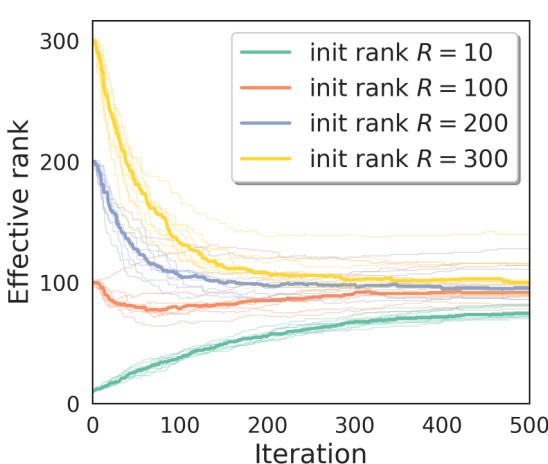

*Figure 4.* Effective CP rank as a function of Gibbs iteration on FFHQ inpainting with the Uniform(0.9) mask, for four initial ranks $R \in \{10, 100, 200, 300\}$. Thick lines are the median across 10 images; thin lines are individual images. Regardless of whether the chain starts from an under-specified ($R = 10$) or over-specified ($R = 300$) initial rank, CUSP drives the effective rank toward a common band of active components.

### D.5. Hyperparameter Sensitivity

Table 5 reports posterior-mean reconstruction quality under perturbations of the coupling constant $c$ and the lower clip $\rho_{\min}$ for DiffBCP, alongside the corresponding sensitivity of PnP-DM (Wu et al., 2024) to its own scheduling and noise hyperparameters.

**DiffBCP is stable across $c$ and $\rho_{\min}$.** Sweeping the coupling constant $c \in \{30, 50, 100\}$ moves Uniform 0.7 PSNR by at most $0.24\,\text{dB}$ and Stripe/Irregular PSNR by at most $0.05\,\text{dB}$. On the most ill-posed mask (Uniform 0.9), $c = 30$ degrades

*Table 5.* Sensitivity to scheduling hyperparameters on FFHQ inpainting (posterior-mean reconstruction). For DiffBCP, the defaults are the coupling constant $c = 100$ and the lower clip $\rho_{\min} = 0.3$; for PnP-DM, the defaults are $\rho_{\min} = 0.1$ and the observation noise $\sigma$ set to the ground-truth value $\sigma = 0.05$. Default configurations are highlighted in green; all other entries change a single hyperparameter and keep the rest at the default value.

| Method | Uniform(0.7) | | | Uniform(0.9) | | | Stripe | | | Irregular | | |
|---|---|---|---|---|---|---|---|---|---|---|---|---|
| | PSNR | SSIM | LPIPS | PSNR | SSIM | LPIPS | PSNR | SSIM | LPIPS | PSNR | SSIM | LPIPS |
| *DiffBCP — varying coupling constant $c$ and lower clip $\rho_{\min}$* | | | | | | | | | | | | |
| DiffBCP ($c = 100$, $\rho_{\min} = 0.3$, default) | 32.13 | 88.83 | 17.70 | 28.28 | 81.18 | 28.93 | 27.91 | 85.11 | 19.89 | 30.34 | 88.60 | 15.98 |
| DiffBCP ($c = 50$) | 31.93 | 88.82 | 17.63 | 27.69 | 80.03 | 29.58 | 27.91 | 85.15 | 19.79 | 30.35 | 88.69 | 15.88 |
| DiffBCP ($c = 30$) | 31.89 | 88.79 | 17.69 | 26.83 | 77.51 | 32.60 | 27.90 | 85.17 | 19.83 | 30.35 | 88.60 | 15.87 |
| DiffBCP ($\rho_{\min} = 0.1$) | 33.04 | 86.80 | 16.22 | 28.31 | 80.90 | 29.64 | 27.71 | 83.92 | 21.61 | 30.01 | 87.66 | 17.39 |
| *PnP-DM — varying lower clip $\rho_{\min}$ and observation noise $\sigma$* | | | | | | | | | | | | |
| PnP-DM ($\rho_{\min} = 0.1$, $\sigma = 0.05$, default) | 33.20 | 88.92 | 17.30 | 28.41 | 81.22 | 25.88 | 27.30 | 86.10 | 17.74 | 30.56 | 89.38 | 14.60 |
| PnP-DM ($\rho_{\min} = 0.3$) | 29.48 | 81.72 | 25.80 | 25.44 | 73.50 | 33.37 | 26.20 | 81.34 | 24.15 | 28.89 | 79.05 | 21.30 |
| PnP-DM ($\sigma = 0.025$) | 33.56 | 89.56 | 16.31 | 28.78 | 82.30 | 24.68 | 27.30 | 86.39 | 17.12 | 30.61 | 89.70 | 13.90 |
| PnP-DM ($\sigma = 0.1$) | 32.05 | 86.78 | 20.37 | 27.07 | 77.60 | 29.73 | 30.31 | 88.21 | 16.75 | 27.20 | 84.99 | 19.74 |

by $1.45$ dB relative to $c = 100$, but no setting catastrophically fails. Reducing $\rho_{\min}$ from the default $0.3$ to $0.1$ actually *improves* Uniform $0.7$ PSNR by $0.91$ dB while losing $0.20$ dB on Stripe and $0.33$ dB on Irregular — a small re-balancing rather than a regime change.

**PnP-DM is more sensitive to scheduling.** By contrast, increasing PnP-DM's $\rho_{\min}$ from the default $0.1$ to $0.3$ causes a sharp collapse: PSNR drops by $3.72$ dB on Uniform $0.7$ and by $2.97$ dB on Uniform $0.9$. The noise-adaptive coupling schedule in DiffBCP, in which $\rho$ is automatically set by the inferred precision $\tau$ (Section 3.2.2), removes the need for delicate manual scheduling of the diffusion noise level.

**PnP-DM benefits from carefully tuned observation noise.** We additionally vary the observation-noise parameter $\sigma$ that PnP-DM passes to its diffusion denoiser. With the ground-truth value $\sigma = 0.05$ already used in the default row, doubling to $\sigma = 0.1$ degrades all four masks (e.g., $-1.15$ dB on Uniform $0.7$), while halving to $\sigma = 0.025$ improves PnP-DM further and matches or surpasses DiffBCP (Full) on most cells (e.g., Uniform $0.7$ PSNR $33.56$ vs. $32.13$). We report this row openly: it confirms that a well-tuned PnP-DM, with $\sigma$ chosen below the true noise to drive more aggressive denoising, can outperform DiffBCP on uniform masks. The point of this comparison is not to claim universal superiority, but to highlight a different operating mode: DiffBCP automatically infers $\tau$ from the data and couples $\rho$ accordingly, so no analogous hand-tuning of $\sigma$ is required. We view the tuning-free regime as the relevant axis for practical deployment, especially in settings where the true noise level is unknown.

### D.6. Out-of-Distribution and High-Resolution Images

**Datasets** We test our algorithm on out-of-distribution high-resolution images. We choose three images with $2048 \times 2048$ resolution, including Marseille[2], Tokyo[3], and Westerlund[4]. We generate the masks in the same way with Section D.1, and use the same iid Gaussian noise with $\sigma = 0.05$ for denoising.

**Implementations** In this experiment, we compare with PuTT (Loeschcke et al., 2024), as it shows SOTA performances for such high-dimensional data. We use the official implementation of PuTT in PyTorch. Additionally, we compare with BCP (Zhao et al., 2015) as a direct comparison to CP decomposition.

For our method, recall that the image size ($2048 \times 2048$) is inconsistent with the training size of the diffusion model ($256 \times 256$). In specific, the reconstructed tensor $\mathcal{X}$ has shape $2048 \times 2048 \times 3$, while the augmented tensor $\mathcal{Z}$ has shape $256 \times 256 \times 3$. Here we use bicubic interpolation for resizing. When compute the coupling distant Eq. (5), we downsample $\mathcal{X}$ to $256 \times 256 \times 3$. When sampling CP factors Eqs. (6) and (7), we upsample $\mathcal{Z}$ back to $2048 \times 2048 \times 3$.

---

[2] https://www.pexels.com/photo/aerial-drone-view-of-urban-buildings-from-top-18644280/
[3] https://www.flickr.com/photos/trevor_dobson_inefekt69/29314390837
[4] https://science.nasa.gov/asset/hubble/westerlund-2/

*Table 6.* Wall-clock time and peak GPU memory per image, measured on a single A100 GPU. FFHQ uses $256 \times 256$ images (Section D.1); Marseille uses $2048 \times 2048$ images (Section D.6).

| Method | **FFHQ** ($256{\times}256$) | | **Marseille** ($2048{\times}2048$) | |
| --- | --- | --- | --- | --- |
| | **Time** | **Peak Mem.** | **Time** | **Peak Mem.** |
| HLRTF | 11.7 s | 593 MB | 137 s | 12.8 GB |
| DeepTensor | 27.3 s | 736 MB | 456 s | 35.3 GB |
| PnP-DM | 37.4 s | 744 MB | 109 s | 5.3 GB |
| DiffBCP | 239 s | 821 MB | 250 s | 12.5 GB |

**Hyperparameters**  PuTT learns the Quantized Tensor Train (QTT) representations in a coarse-to-fine manner. We set the downsampling resolution as $[64, 128, 256, 512, 1024, 2048]$, which means that the image is downsampled to $64 \times 64$ at the coarsest level and upsampled in sequence. The total number of iterations is set as 2560, and we upsample the image at $64, 128, 256, 512, 1024$ iterations. The learning rate is $8\mathrm{e}{-}3$ using Adam optimizer and batch size 262144. The maximum QTT rank is set as 100.

For BCP and our method, we reshape the images into $64 \times 32 \times 64 \times 32 \times 3$ tensors. For BCP, we set the initial CP rank as 100, because it quickly exceeds the maximum memory limit (512 GB) when the rank is larger. Besides, even with rank 100, we find BCP quickly shrinks the effective rank. For our method, we set the initial CP rank as 500. Other hyperparameters are the same as in Section D.1. Note that due to the different rank definition of QTT and CP, our CP format still yields larger compression rate than QTT.

**Evaluation**  The evaluation is the same as in Section D.1. We run each experiment with five different random seeds and report the averaged results.

### D.7. Computational Cost

Table 6 reports wall-clock time and peak GPU memory per image on a single A100 GPU, for the FFHQ $256 \times 256$ setup of Section D.1 and the Marseille $2048 \times 2048$ setup of Section D.6.

DiffBCP is slower than PnP-DM, reflecting the cost of running full-batch CP factor updates. In the high-resolutional Marseille case, DiffBCP runs in 250 s with 12.5 GB peak memory, which is $1.8\times$ faster and $2.8\times$ more memory-efficient than the strongest deep tensor baseline DeepTensor (456 s, 35.3 GB).

The main bottleneck of DiffBCP is the full-batch Gibbs update on $\mathbf{A}$ and $\boldsymbol{\lambda}$. As noted in the conclusion, replacing this with stochastic mini-batch sampling is a natural direction to further reduce runtime and memory, particularly when scaling to even larger tensors.

### D.8. More Visualizations

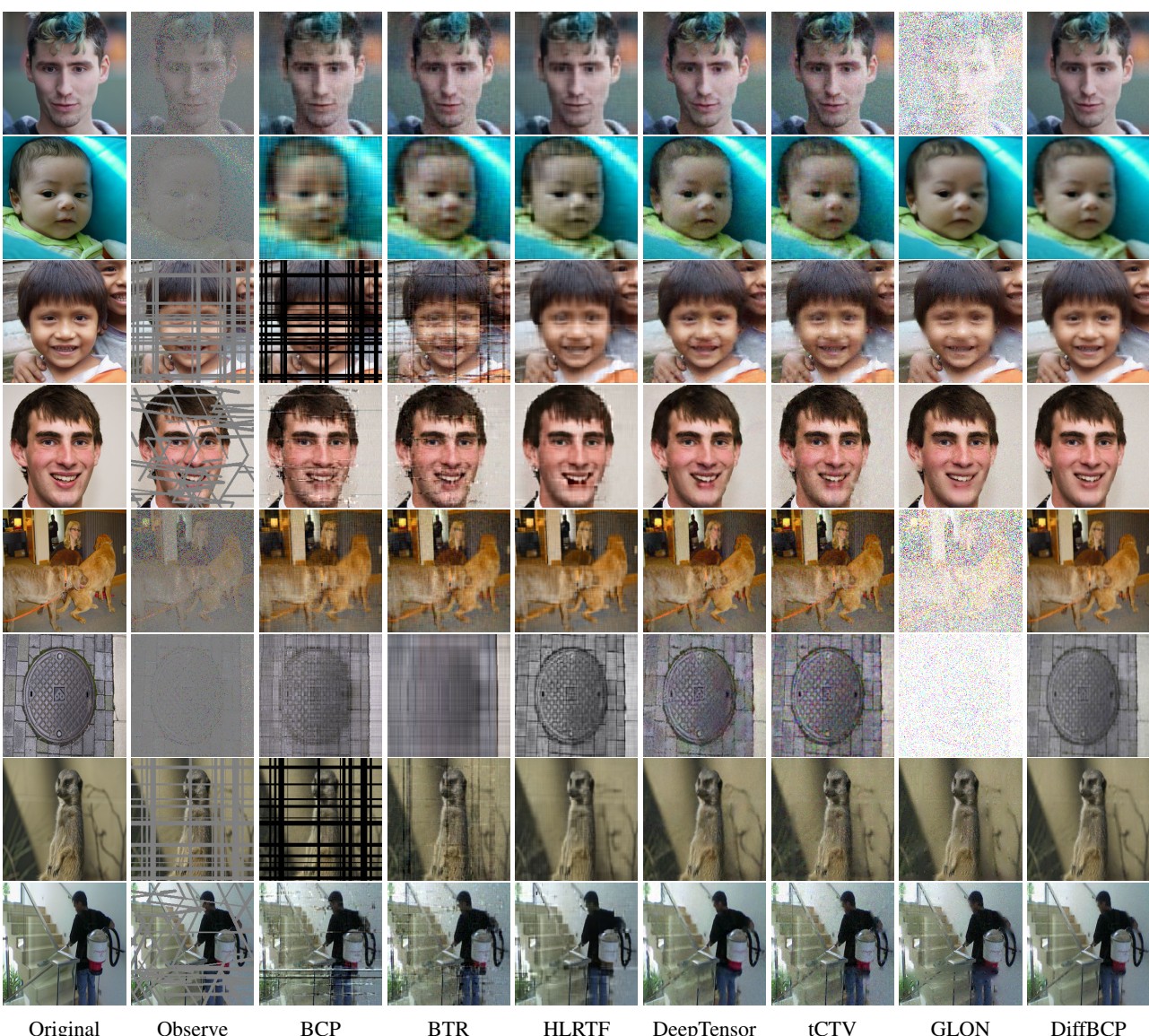

| Original | Observe | BCP | BTR | HLRTF | DeepTensor | tCTV | GLON | DiffBCP |

*Figure 5.* More visualization for FFHQ and ImageNet datasets.

