# OpenReview forum: "Bayesian Tensor Decomposition with Diffusion Model Prior"
_ICML.cc/2026/Conference — ICML 2026 regular_

### Official Review · Reviewer_ntG4 · 2026-03-10

**Soundness:** 4
**Presentation:** 3
**Significance:** 4
**Originality:** 4
**Overall Recommendation:** 5
**Confidence:** 5

**Summary:**

This paper proposes a novel Bayesian tensor decomposition framework named DiffBCP. Its core innovation lies in incorporating a pre-trained diffusion model as an implicit prior into traditional Bayesian CP decomposition to address the expressive limitations of conventional low-rank constraints under extremely high missing rates or severe noise. The authors employ a split Gibbs sampling noise adaptation coupling scheme to simplify hyperparameter tuning. Experiments demonstrate that DiffBCP significantly outperforms conventional methods in image completion and denoising tasks, while exhibiting strong robustness when handling out-of-distribution and high-resolution images.

**Compliance With Llm Reviewing Policy:**

Affirmed.

**Key Questions For Authors:**

1. How is the number of steps for SDE sampling determined in each Gibbs iteration? Is c=50 universally applicable? Additional experiments on sensitivity to c values are needed.
2. When processing 2048x2048 high-resolution images, since the diffusion model is pre-trained on 256x256, does using bicubic interpolation for size mapping cause loss of high-frequency information?
3. Explaining how parameter initialization values are determined, particularly how Z is initialized, would significantly enhance understanding of the paper.

**Limitations:**

yes

**Strengths And Weaknesses:**

Strengths
- Methodological Innovation: Integrating diffusion models into the Bayesian tensor framework represents a highly promising direction. Embedding SDE sampling into the Gibbs loop via SGS not only preserves the conjugacy of CP factor updates but also endows the low-rank model with the capability to capture complex high-dimensional distributions.
- Automatic Rank Determination: Theorem 3.1 demonstrates that the CUSP prior effectively contracts residuals in CP decompositions, providing theoretical justification for automatic rank selection. Combining CUSP prior-based rank selection with an adaptive scheme that automatically infers coupling parameter ρ via noise precision τ significantly lowers the tuning threshold for such complex models in practical applications.
- Generalization Capability: Experiments demonstrate the model's outstanding performance under 90% missing data rates and OOD high-resolution images (e.g., 2048x2048), particularly in preserving global consistency and fine-grained texture compared to methods like PuTT.

Weaknesses
- When processing 2048×2048 images, bicubic interpolation is used to downsample X to Z's space to fit the 256×256 pre-trained diffusion model, followed by upsampling back to the original size. The “details” recovered by the model primarily rely on the linear interpolation capability of CP decomposition or the diffusion model's “generation” at low resolutions.

---

> ### Author Rebuttal · Authors · 2026-03-31
>
> Thank you for your strong support and insightful questions.
>
> **Q1: How is the number of steps for SDE sampling determined in each Gibbs iteration? Is $c=50$ universally applicable?**
> **A1:** In our implementation, after sampling precision $\tau_i$, we compute the coupling parameter $\rho_i = \sqrt{c/\tau_i}$ (clipped to a predefined range). We then match the pretrained diffusion noise schedule to find a starting time step $T_i$ such that $\sigma(T_i) \approx \rho_i$. Thus, the number of SDE denoising steps is dynamically determined by $\rho_i$; larger $\rho_i$ yields more denoising steps. The value $c=50$ is our empirical default rather than a universal constant. We will add a sensitivity study for $c$ (as requested by Reviewer 83Ex) to illustrate the trade-off.
>
> **Q2: When processing 2048x2048 images, does using bicubic interpolation for size mapping cause loss of high-frequency information?**
> **A2:** Yes, this creates a genuine information bottleneck. Because the auxiliary variable $Z$ is sampled in the downsampled 256x256 space, the diffusion prior primarily regularizes global semantics and low/mid-frequency structure. High-frequency details are instead recovered by the overarching observation likelihood and the high-resolution CP structure imposed on $X$ in the original 2048x2048 space. We will calibrate our claims in Sec. 4.2 appropriately and provide control visualizations to show the detail recovery is not strictly attributable to simple interpolation.
>
> **Q3: Explain how parameter initialization values are determined, particularly how $Z$ is initialized.**
> **A3:** We will update Algorithm 1 and the appendix to explicitly detail the initialization. Specifically, $Z$ is initialized from the initial reconstruction tensor $X$ and subsequently updated by the diffusion SDE step in each succeeding Gibbs iteration. We will also include an ablation comparing reconstruction-based initialization against random noise initialization for $Z$, confirming our method remains robust to initialization choice.

---

> > ### Author Rebuttal · Reviewer_ntG4 · 2026-04-04
> >
> > The authors’ responses have clarified my questions. Accordingly, I retain my original score.

---

> > > ### Author Response · Authors · 2026-04-06
> > >
> > > Thanks very much for your support. We will revise the manuscript accordingly to reflect all the discussions above.

---

### Official Review · Reviewer_83Ex · 2026-03-11

**Soundness:** 2
**Presentation:** 2
**Significance:** 2
**Originality:** 3
**Overall Recommendation:** 3
**Confidence:** 3

**Summary:**

The paper proposes DiffBCP, a Bayesian CP tensor decomposition framework that combines a cumulative shrinkage process (CUSP) prior for automatic rank selection with an off-the-shelf pretrained diffusion model as an implicit prior over the reconstructed tensor. The authors develop a split Gibbs sampler with an auxiliary variable to decouple the likelihood, low-rank constraint, and diffusion prior, and they introduce a noise-adaptive coupling schedule that ties the coupling strength to the inferred noise precision. Empirically, DiffBCP is evaluated on image inpainting and denoising for FFHQ and ImageNet, as well as high-resolution out-of-distribution images, showing improvements over several Bayesian, deep, and plug-and-play tensor decomposition baselines in PSNR, SSIM, and LPIPS. The paper also includes theoretical analysis of the bias induced by the smoothed diffusion prior and inexact denoising steps in the split Gibbs sampler.

**Compliance With Llm Reviewing Policy:**

Affirmed.

**Key Questions For Authors:**

1. Can you provide an ablation where the diffusion prior is removed (reducing DiffBCP to a CUSP-based Bayesian CP model) and, conversely, where the tensor decomposition is removed but the same diffusion prior is used with the split Gibbs sampler? This would clarify how much each component contributes to the reported gains.​
2. How does the effective CP rank evolve during sampling across different datasets and mask patterns, and how sensitive is performance to the CUSP hyperparameters and the rank adaptation schedule? Providing plots of effective rank vs. iterations and a small hyperparameter sweep would make the automatic rank selection claim more convincing.​
3. Can you add quantitative PSNR/SSIM/LPIPS comparisons against PnP-DM and at least one other recent diffusion posterior sampler on the same inpainting/denoising benchmarks? This would better support the claim that incorporating low-rank tensor structure improves over diffusion-only approaches.​
4. What are the typical wall-clock times and GPU memory footprints for DiffBCP compared to HLRTF, DeepTensor, GLON, and PnP-DM on the 256×256 and 2048×2048 experiments? Without such numbers, it is difficult to judge whether the method is practical for large-scale real-world deployments.​
5. Can you design experiments that explicitly vary the coupling constant cc and clipping range of ρρ, and relate the observed reconstruction performance and mixing behavior to the theoretical bias bounds in Theorem 3.4 and Corollary 3.5? Such experiments would significantly strengthen the connection between the theory and practice.

*If Concerns are solved, I will raise my rating.

**Limitations:**

yes

**Strengths And Weaknesses:**

Strengths:
1. The integration of a CUSP prior with a diffusion-model-based implicit prior in a unified Bayesian CP framework is conceptually appealing and technically nontrivial.

2. The split Gibbs sampler with an auxiliary variable and a noise-adaptive coupling schedule is a principled attempt to make posterior inference tractable despite the implicit diffusion prior.

3. Experimental results on both standard-scale and high-resolution OOD images are strong in terms of reconstruction metrics and visual quality, outperforming a broad set of tensor-decomposition baselines.


Weakness:
1. The paper does not systematically ablate the diffusion prior, the CUSP‑based rank adaptation, or the noise schedule, so it is unclear how much each design choice actually contributes to the reported gains.
​

2. Insufficient comparison to strong diffusion posterior samplers
Although the method is framed as related to PnP‑DM and other diffusion posterior sampling approaches, the empirical comparison is narrow and mostly qualitative, without clear PSNR/SSIM/LPIPS comparisons on shared benchmarks.
​

3. The paper does not study behavior on clearly non‑low‑rank data, nor does it show how the effective rank evolves or how sensitive performance is to CUSP hyperparameters and adaptation schedules, so the “automatic rank selection and robustness” claim is not fully substantiated.
​

4. Combining MCMC over tensor factors with repeated diffusion denoising is likely much more expensive than traditional TD or some diffusion samplers, yet the paper reports no wall‑clock time, memory usage, or cost comparison, making the practical trade‑off between quality and efficiency unclear.

---

> ### Author Rebuttal · Authors · 2026-03-31
>
> Thank you for the detailed feedback. We have added the required ablations, comparisons, and measurements.
>
> **Q1: Ablation: diffusion prior is removed, and tensor decomposition is removed.**
> **A1:** The diffusion prior ablation is provided in **our response to Reviewer LCUU**, confirming its substantial gains on structured masks. Conversely, removing the tensor decomposition reduces our method to standard diffusion posterior sampling. A comprehensive comparison against **PnP-DM** and **DPS** (see **Reviewer tUPg's response**) shows that our structural low-rank CP constraint yields superior single-sample quality and competitive posterior means versus diffusion-only samplers.
>
> **Q2: How does the effective CP rank evolve across different datasets, and how sensitive is performance to CUSP hyperparameters?**
> **A2:** We will include active rank trajectory plots in the revision. In principle, given sufficient Markov chain iterations, the CUSP prior automatically converges to an optimal effective rank. To demonstrate this, we initialized our model with an under-specified rank (**see Reviewer LCUU**): while static fixed-rank models fail severely, our CUSP-enabled model dynamically recovers performance. Additionally, our framework is largely tuning-free. By employing standard, weakly-informative priors (e.g., $\mathrm{Beta}$, $\mathrm{InvGamma}$ with default constants), shrinkage severity is inferred directly from the data posterior, eliminating the need for dataset-specific tuning.
>
> **Q3: Quantitative PSNR/SSIM/LPIPS comparisons against diffusion posterior sampler?**
> **A3:** Yes. Please refer to **our response to Reviewer tUPg** for matched quantitative comparisons against PnP-DM and DPS.
>
> **Q4: Wall-clock times and GPU memory footprints for DiffBCP compared to baselines on 256x256 and 2048x2048 experiments?**
> **A4:** We measured runtime and peak memory, tracking the cost of repeated SDE denoising combined with CP latent updates. We do not compare with GLON as it is nontrival to adapt the Matlab code to our GPU. Experiments are conducted on Nvidia A100.
>
>
> |Method|FFHQ(256)Time|PeakMem.|Marseille(2048)Time|PeakMem.|
> |---|---|---|---|---|
> |HLTRF|11.66s|593MB|137s|12809MB|
> |DeepTensor|27.33s|736MB|456s|35328MB|
> |PnP-DM|37.35s|744MB|109s|5294MB|
> |DiffBCP|239s|821MB|250s|12543MB|
>
> While DiffBCP requires more compute than standard baselines on $256\times 256$ images due to the concurrent diffusion SDE and tensor updates, it scales more favorably to $2048\times 2048$ resolutions compared to nonlinear deep TD methods in runtime, as their nonlinear structures may be extremely complex for large tensors. Specifically compared to PnP-DM, our additional computational cost primarily stems from the BCP block. Our current BCP implementation utilizes full-batch Gibbs sampling, which can be time-consuming. To address this, we may adopt stochastic mini-batch sampling algorithms to significantly accelerate this step in the future, as we have discussed in Section 5.
>
> **Q5: Can you design experiments that explicitly vary the coupling constant \(c\) and clipping range of \(\rho\)?**
> **A5:** We have designed a targeted sensitivity study varying the coupling constant $c$ (which scales the dynamically inferred $\rho$) and the threshold $\rho_{\min}$.
>
> |Method|Uniform(0.7) PSNR↑|SSIM↑|LPIPS↓|Uniform(0.9) PSNR↑|SSIM↑|LPIPS↓|Stripe PSNR↑|SSIM↑|LPIPS↓|Irregular PSNR↑|SSIM↑|LPIPS↓|
> |---|---|---|---|---|---|---|---|---|---|---|---|---|
> |DiffBCP(c=100)|32.13|88.83|17.70|28.28|81.18|28.93|27.91|85.11|19.89|30.34|88.60|15.98|
> |DiffBCP(c=50)|31.93|88.82|17.63|27.69|80.03|29.58|27.91|85.15|19.79|30.35|88.69|15.88|
> |DiffBCP(c=30)|31.89|88.79|17.69|26.83|77.51|32.60|27.90|85.17|19.83|30.35|88.60|15.87|
> |DiffBCP($\rho_{\min}=0.1$)|33.04|86.80|16.22|28.31|80.90|29.64|27.71|83.92|21.61|30.01|87.66|17.39|
> |PnP-DM($\rho_{\min}=0.1$)|33.20|88.92|17.30|28.41|81.22|25.88|27.30|86.10|17.74|30.56|89.38|14.60|
> |PnP-DM($\rho_{\min}=0.3$)|29.48|81.72|25.80|25.44|73.50|33.37|26.20|81.34|24.15|28.89|79.05|21.30|
>
> As shown, varying $c \in \{30, 50, 100\}$ yields remarkably stable performance. Furthermore, compared to a pure diffusion sampler (PnP-DM) which suffers a severe performance collapse when $\rho_{\min}$ increases to $0.3$, our low-rank Bayesian CP structure intrinsically stabilizes the inference process, suppressing this extreme sensitivity to diffusion hyperparameter tuning.
>
> **Q6: Behavior on clearly non-low-rank data or failure modes?**
> **A6:** Theoretically, CP decomposition can approximate arbitrary tensors given sufficient rank, and CUSP dynamically adjusts this effective rank, preventing abrupt "failures". However, if the underlying data lacks meaningful low-rank structure, the structural benefits of CP diminish. In such extreme high-rank cases, the computational overhead of updating massive factor matrices makes our hybrid approach less advantageous than pure diffusion methods. We will clarify these boundary conditions.

---

> > ### Author Rebuttal · Reviewer_83Ex · 2026-04-04
> >
> > Thank you for the comprehensive response and for clarifying how the method behaves on data that lacks meaningful low-rank structure. Acknowledging this boundary condition is helpful, but I have a short follow-up question about how users might detect this extreme high-rank regime in practice.

---

> > > ### Author Response · Authors · 2026-04-06
> > >
> > > Thank you for the helpful follow-up. We would like to clarify that the main advantage of our CUSP-based rank adaptation is that it improves **robustness to rank misspecification across regimes**: when the underlying signal is effectively low-rank, CUSP prunes redundant components and avoids over-parameterization; when a fixed low rank is insufficient, it can activate additional components and prevent underfitting. In this sense, automatic rank selection is a core strength of DiffBCP, because it removes the need to accurately specify the CP rank in advance.
> > >
> > > This behavior is consistent with our ablation results. When we initialize the model with an under-specified rank, the fixed-rank version degrades substantially, whereas the CUSP-enabled version remains much more robust. We view this as direct evidence that the benefit of automatic rank selection is precisely to adapt the model complexity to the data, rather than relying on a carefully tuned rank chosen beforehand.
> > >
> > > A practical sign that the data may have entered an extreme high-rank regime is that the effective rank keeps increasing. In that case, the posterior is indicating that many components need to remain active. We will add this practical guidance to the discussion.
> > > Furthermore, we will revise the manuscript to emphasize a more accurate claim: one key contribution of CUSP is to make DiffBCP adaptive to unknown rank, yielding both automatic pruning in simpler settings and automatic expansion in more complex ones, instead of requiring a manually tuned fixed rank.

---

### Official Review · Reviewer_LCUU · 2026-03-16

**Soundness:** 3
**Presentation:** 3
**Significance:** 3
**Originality:** 3
**Overall Recommendation:** 5
**Confidence:** 4

**Summary:**

Considering that low-rank tensor decomposition (TD) degrades under severe missingness or noise, this work proposes DiffBCP, a Bayesian CP decomposition framework that combines a cumulative shrinkage process prior for automatic rank selection with a pre-trained diffusion model as an implicit data-driven prior. Then, a split Gibbs sampler is developed to enable tractable posterior inference, while the diffusion block is sampled via low-rank-guided denoising. A series of experiments on image inpainting and denoising tasks are carried out to show consistent gains over several baselines.

**Compliance With Llm Reviewing Policy:**

Affirmed.

**Final Justification:**

The authors have addressed my concerns.

**Key Questions For Authors:**

1.  How does the proposed procedure generalize from CP decomposition to other tensor decompositions, such as tensor train or tensor ring?

2. What criteria guide the selection of pre-trained diffusion models in practice?

**Limitations:**

Yes.

**Strengths And Weaknesses:**

Strengths
1. A new CP decomposition that combines a cumulative shrinkage process prior and pre-trained diffusion model priors is proposed to capture complex data distributions. This seems to be novel in the literature.

2. The proposed framework employs fully probabilistic inference, offering very strong adaptability and generalization.

3. Both theoretical analysis and numerical experiments are conducted to show the merits of the proposed framework.

Weaknesses:

1. This work lacks ablation studies.

2. The proof techniques of Theorem 3.4 seems to follow directly from Heurtel-Depeiges et al. (2024), with limited originality.

---

> ### Author Rebuttal · Authors · 2026-03-31
>
> Thank you for the positive assessment of our framework's novelty and technical solidity.
>
> **Q1: The work lacks ablation studies.**
> **A1:** We have conducted comprehensive ablations isolating the diffusion prior and the CUSP mechanism. The results demonstrate that:
> 1. The **diffusion prior** significantly boosts performance on structured masks (Stripe/Irregular).
> 2. The **CUSP mechanism** provides robustness, enabling strong recovery even when the initial rank is misspecified.
>
> | Method | Uniform(0.7) PSNR↑ | SSIM↑ | LPIPS↓ | Uniform(0.9) PSNR↑ | SSIM↑ | LPIPS↓ | Stripe PSNR↑ | SSIM↑ | LPIPS↓ | Irregular PSNR↑ | SSIM↑ | LPIPS↓ |
> |---|---|---|---|---|---|---|---|---|---|---|---|---|
> | DiffBCP (Full) | 32.13 | 88.83 | 17.70 | 28.28 | 81.18 | 28.93 | 27.91 | 85.11 | 19.89 | 30.34 | 88.60 | 15.98 |
> | DiffBCP (w/o DM) | 32.05 | 88.00 | 18.88 | 28.10 | 79.24 | 32.52 | 25.82 | 79.35 | 28.97 | 28.64 | 85.28 | 21.39 |
> | DiffBCP (rank=10, w/ CUSP) | 31.62 | 88.02 | 18.97 | 28.26 | 80.80 | 29.75 | 27.22 | 82.25 | 23.60 | 29.34 | 86.15 | 19.50 |
> | DiffBCP (rank=10) | 26.35 | 75.17 | 37.39 | 25.83 | 73.42 | 40.75 | 23.60 | 68.80 | 44.17 | 25.23 | 72.68 | 40.64 |
>
> **Q2: How does the proposed procedure generalize from CP decomposition to other tensor decompositions, such as tensor train (TT) or tensor ring (TR)?**
> **A2:** The core of DiffBCP is decomposition-agnostic: the observation model, the diffusion block for \(Z\), and the split Gibbs logic remain unchanged. To extend to TT/TR, one only replaces the CP reconstruction map with the TT/TR contraction. Since each core in TT/TR enters the reconstruction linearly (conditioned on the other cores), substituting Gaussian priors ensures the conditional posterior of each core remains a tractable Gaussian block update. We will clarify this generalizability and sketch a TT/TR update in the appendix.
>
> **Q3: What criteria guide the selection of pre-trained diffusion models in practice?**
> **A3:** We will add the following practical guidance for prior selection:
> 1. **Modality/Domain match:** Always prefer models trained on semantically similar domains.
> 2. **Compatibility:** Ensure pre-processing (normalization, channels) and resolution compatibility.
> 3. **General vs. Narrow:** If an exact domain match is unavailable, a broader general prior is preferable to a highly specialized but mismatched one.
>
> **Q4: The proof techniques of Theorem 3.4 seem to follow directly from Heurtel-Depeiges et al. (2024), with limited originality.**
> **A4:** We agree and will make the attribution to Heurtel-Depeiges et al. (2024) more prominent. Our theoretical contribution is not the underlying proof machinery, but rather correctly applying this stationary-bias analysis to our unique augmented posterior $\pi_\rho(Z,\Theta)$, where $\Theta$ captures the structured Bayesian tensor block. We will revise the text to ensure our theoretical novelty claims are appropriately bounded.

---

> > ### Author Rebuttal · Reviewer_LCUU · 2026-04-02
> >
> > I think the authors have addressed my concerns. As such, I have raised my score to 5.

---

> > > ### Author Response · Authors · 2026-04-06
> > >
> > > Thanks very much for your support and raising up your score. We will revise the manuscript accordingly to reflect all the discussions above.

---

### Official Review · Reviewer_tUPg · 2026-03-20

**Soundness:** 3
**Presentation:** 3
**Significance:** 2
**Originality:** 2
**Overall Recommendation:** 3
**Confidence:** 4

**Summary:**

The authors propose a framework for solving inverse problems (such as denoising/inpainting) with a Bayesian framework using the inductive bias that the reconstruction should be well-approximated with a low-rank CP decomposition. The authors rely on a diffusion prior to capture the likelihood of the reconstruction and use other hand-crafted probabilistic priors for the likelihood of the measurements given the reconstruction and for the likelihood of the factor matrices. The authors show that using a diffusion prior instead of a hand-crafted one gives superior results compared to prior Bayesian methods across various image reconstruction tasks.

**Compliance With Llm Reviewing Policy:**

Affirmed.

**Key Questions For Authors:**

See weaknesses above. I would like to see how this method compares to state-of-the-art algorithms for solving inverse problems with diffusion models.

**Limitations:**

N/A.

**Strengths And Weaknesses:**

Strengths:

- The problem of solving inverse problems with diffusion priors is super relevant and important.
- The idea of low-rank CP regularization for the solution of inverse problems is novel.
- The integration of the diffusion prior as a regularizer is more powerful than hand-crafted regularizations. This is intuitive but the authors also show it experimentally.
- The authors propose a novel sampling method and show interesting theoretical results regarding its convergence to the true measure.


Weaknesses:

- The proposed method is not compared against other methods for solving inverse problems with diffusion models, such as DPS, SNIPS, DDRM, and other more modern variants.
- Visually, the results are better than the baselines, but still not super impressive compared to modern editing methods. Currently, in my opinion, there isn't a very compelling argument as to why this method is preferable to other methods in the huge literature of solving inverse problems with diffusion priors.
- The idea of replacing handcrafted priors with generative priors has a very long history, starting from the seminal CSGM paper. I think that the paper would benefit from a discussion of this history, as currently the discussion is mostly limited to the Bayesian TD literature.
- In some sense, the low-rank decomposition assumption is a "hand-crafted" assumption. This, in my opinion, is a little bit at odds with the story of the paper, which emphasizes moving towards data-driven priors.
- The authors analyze the measure at which their sampling algorithm will converge, but do not provide any theoretical analysis of (i) the time it takes for convergence and (ii) the propagation of errors from the estimation of the score function.
- To run this method, one needs diffusion priors trained on *clean* data. This prohibits the application of the method to cases when clean data is not available in the first place.

---

> ### Author Rebuttal · Authors · 2026-03-31
>
> Thank you for your constructive feedback.
>
> **Q1: Not compared against other methods for solving inverse problems with diffusion models.**
> **A1:** We agree and have added matched comparisons against **DPS** and **PnP-DM** on the FFHQ dataset inpainting benchmark. Below we report the posterior-mean and single-sample reconstruction performance.
> |Method|Uniform(0.7)PSNR↑|SSIM↑|LPIPS↓|Uniform(0.9)PSNR↑|SSIM↑|LPIPS↓|StripePSNR↑|SSIM↑|LPIPS↓|IrregularPSNR↑|SSIM↑|LPIPS↓|
> |---|---|---|---|---|---|---|---|---|---|---|---|---|
> |DPS|30.75|87.31|15.61|26.98|78.81|21.22|27.37|85.27|14.84|29.87|88.70|12.67|
> |PnP-DM(single)|29.45|70.87|24.65|27.21|65.30|31.91|25.82|68.20|25.42|28.04|71.04|22.79|
> |DiffBCP(single)|31.49|85.94|21.31|27.80|77.24|33.67|27.51|81.36|23.93|29.84|85.53|19.71|
> |PnP-DM(mean)|33.20|88.92|17.30|28.41|81.22|25.88|27.30|86.10|17.74|30.56|89.38|14.60|
> |DiffBCP(mean)|32.13|88.83|17.70|28.28|81.18|28.93|27.91|85.11|19.89|30.34|88.60|15.98|
>
> Under posterior-mean evaluation, our DiffBCP is competitive with PnP-DM across all masks, and outperforms DPS in most settings. Under single-sample evaluation, DiffBCP consistently outperforms PnP-DM, indicating that the low-rank constraint effectively stabilizes and improves individual posterior samples. We have partly discussed this phenomeneon in Section 4.3; we will update the manuscript to reflect these results and clarify that our method provides complementary structural advantages rather than universal superiority over diffusion-only samplers.
>
> **Q2: Visually, the results are better than baselines but not super impressive compared to modern editing methods. Why is this method preferable?**
> **A2:** The primary advantage of DiffBCP is fusing an explicit low-rank CP structure, CUSP-based rank shrinkage, and a diffusion prior within a principled Bayesian framework. This combination is particularly crucial for highly ill-posed inverse problems (e.g., extreme missingness or structured corruption), where diffusion-only methods operate in a largely unconstrained space and can hallucinate or fail. DiffBCP structurally regularizes the solution, complementing the semantic prior of the diffusion model.
>
> **Q3: The idea of replacing handcrafted priors with generative priors has a long history (e.g., CSGM).**
> **A3:** We fully agree. Extending Section 1 and 1.2, we will explicitly position DiffBCP within the trajectory of generative priors for inverse problems: progressing from handcrafted priors to generator-based priors (CSGM), Deep Image Prior, PnP/RED, and recent score/diffusion-based methods. Importantly, DiffBCP is a *hybrid* model coupling a tractable structural prior (Bayesian CP) with an expressive data-driven prior (diffusion).
>
> **Q4: The low-rank decomposition assumption is "hand-crafted", which contradicts the data-driven prior story.**
> **A4:** This is a fair point. We will correct our framing to clarify that low-rank CP is indeed a handcrafted structural prior. DiffBCP is a **hybrid-prior** model: the CP/CUSP structure provides compactness and tractable Bayesian inference, while the diffusion model provides rich natural-image statistics. It augments, rather than completely replaces, structural assumptions.
>
> **Q5: The theory analyzes the limiting measure but lacks analysis of convergence time and error propagation from score estimation.**
> **A5:** We agree that analyzing non-asymptotic convergence and error propagation is valuable. However, establishing rigorous mixing bounds for diffusion-based MCMC remains an open challenge. As our core contributions are methodological and empirical, a full theoretical characterization of convergence rates is beyond our current scope. We will explicitly state this limitation and instead provide robust *empirical* convergence diagnostics (e.g., trace plots, autocorrelation, reconstruction stabilization curves) to demonstrate reliable convergence behavior.
>
> **Q6: Running this method requires diffusion priors trained on clean data, limiting its applicability.**
> **A6:** We acknowledge this and will explicitly state that DiffBCP depends on a pre-trained generative model. However, our reliance on exact domain matching is not overly restrictive. Our cross-domain experiments (e.g., applying an ImageNet-trained prior to FFHQ) show only a surprisingly small performance gap between matched and mismatched priors. This suggests the diffusion model primarily provides general low/mid-frequency regularization, while the explicit CP decomposition adapts to specific image structures. We will include these quantitative results and practical guidelines for prior selection.

---

> > ### Author Rebuttal · Reviewer_tUPg · 2026-03-31
> >
> > The authors acknowledged the limitations of their method. I believe that the paper will benefit from a more extensive discussion of the literature, more comparisons with (approximate) posterior sampling algorithms (and improved versions of DPS, which is now 4 years old), and the clarifications the authors promised above.
> >
> > I will slightly raise my score to indicate that I appreciate the honesty of the authors and the commitment to improve their paper.

---

> > > ### Author Response · Authors · 2026-04-06
> > >
> > > Thank you again for the thoughtful follow-up and raising your score.
> > >
> > > **More extensive discussion of the literature**
> > >
> > > We agree that the paper should be positioned more carefully within the broader literature on generative priors and diffusion-based posterior inference. In the revision, we will expand the introduction and related work sections to better connect our method to the progression from classical handcrafted priors, to generator-based priors such as CSGM, to PnP/RED methods, and more recent score-/diffusion-based posterior samplers (DPS, SNIPS, DDRM, etc.).
> > >
> > > We also agree that the low-rank CP assumption is itself a handcrafted structural prior. We will therefore revise the framing to avoid suggesting that our method replaces handcrafted priors with learned ones. A more accurate characterization is that DiffBCP is a **hybrid-prior Bayesian model**: CP/CUSP contributes explicit structural regularization and tractable posterior updates, while the diffusion model contributes a learned natural-image prior.
> > >
> > > **More comparisons with posterior sampling algorithms**
> > >
> > > Regarding comparisons to diffusion posterior samplers, we agree this is an important axis. At the same time, we would like to clarify the intended scope of the paper. Our goal is not to claim universal superiority over diffusion-only posterior samplers. Rather, our claim is that adding explicit low-rank Bayesian structure provides a complementary advantage, especially in severely ill-posed settings where diffusion-only samplers operate in a less constrained posterior space. In our added experiments, DiffBCP is competitive with PnP-DM/DPS in posterior-mean reconstruction and shows a clearer advantage in single-sample reconstruction.
> > >
> > > We would also like to note that PnP-DM is already a relatively strong and representative diffusion baseline for this comparison. Recent large-scale benchmarking work (e.g., [_InverseBench_](https://devzhk.github.io/InverseBench/)) evaluates 14 plug-and-play diffusion-prior algorithms and shows that methods such as PnP-DM tend to be stronger than simpler samplers like DPS in settings with tractable forward-model access, albeit with higher tuning and computational cost. This is why we believe PnP-DM is an informative baseline for our setting.
> > >
> > > Overall, we will revise the paper to make these points clearer: (i) DiffBCP should be positioned as a hybrid structural + learned-prior method, (ii) our contribution is a complementary structural advantage rather than universal dominance over diffusion-only samplers, (iii) the broader generative-prior and diffusion-posterior-sampling literature should be discussed and compared more explicitly and (iv) the empirical comparisons with diffusion posterior sampling algorithms.

---

### Decision · Program_Chairs · 2026-04-30

**Decision:**

Accept (regular)

**Comment:**

The paper proposes a Bayesian CP decomposition framework that combines a cumulative shrinkage process prior for automatic rank determination with a pre-trained diffusion model as an implicit data-driven prior, together with a split Gibbs sampler for tractable posterior inference. The diffusion component is sampled through a low-rank-guided denoising procedure. The paper also provides a rich set of theoretical results, including guarantees related to rank selection and approximation error.

Overall, this is a solid and well-motivated piece of work that opens an interesting new direction for Bayesian tensor decomposition. The paper integrates classical Bayesian inference via Gibbs sampling, nonparametric shrinkage priors, and modern generative modeling in a coherent and technically meaningful way. It has the potential to inspire follow-up work in tensor decomposition and related areas.

Some reviewers raised concerns about performance in high-rank regimes. However, this issue is somewhat less central in the tensor decomposition setting, which is primarily concerned with discovering low-rank structure. That said, the authors would still benefit from addressing the other reviewer concerns more carefully, in particular by providing a more extensive discussion of the related literature.